# Chromatin information content landscapes inform transcription factor and DNA interactions

Ricardo D'Oliveira Albanus [1], Yasuhiro Kyono[1,2,3], John Hensley[1], Arushi Varshney [1], Peter Orchard[1], Jacob O. Kitzman [1,2] & Stephen C. J. Parker [1,2✉]

Interactions between transcription factors and chromatin are fundamental to genome organization and regulation and, ultimately, cell state. Here, we use information theory to measure signatures of organized chromatin resulting from transcription factor-chromatin interactions encoded in the patterns of the accessible genome, which we term chromatin information enrichment (CIE). We calculate CIE for hundreds of transcription factor motifs across human samples and identify two classes: low and high CIE. The 10–20% of common and tissue-specific high CIE transcription factor motifs, associate with higher protein–DNA residence time, including different binding site subclasses of the same transcription factor, increased nucleosome phasing, specific protein domains, and the genetic control of both chromatin accessibility and gene expression. These results show that variations in the information encoded in chromatin architecture reflect functional biological variation, with implications for cell state dynamics and memory.

[1] Department of Computational Medicine & Bioinformatics, University of Michigan, Ann Arbor, USA. [2] Department of Human Genetics, University of Michigan, Ann Arbor, USA. [3] Present address: Tempus Labs, Inc. Chicago, IL Chicago, USA. ✉email: scjp@umich.edu

Understanding the interactions between transcription factors (TFs) and chromatin is critical to dissect regulatory circuits that lead to differences in transcriptional activity across species, tissues, stimulatory, and genetic contexts. Chromatin is the association between DNA, RNA, and diverse nuclear proteins, including nucleosomes. It enables the ~2-meter human genome to be packaged inside the nucleus while allowing active genes and their corresponding regulatory elements to remain accessible[1]. Nucleosome positioning is an essential property of chromatin architecture and has been shown to have both passive and active roles in TF binding[2–4]. Information theory provides a powerful framework to quantify ordered patterns in data[5] and has been successfully used to characterize genome-wide DNA methylation patterns[6].

In this work, we develop information-theoretical tools to study TF-chromatin interactions in human tissues using chromatin accessibility data. We show that local chromatin architecture encodes information-rich signatures of TF interactions. Our results show that variations in the information patterns encoded in chromatin architecture reflect functional biological variation, with implications for cell state dynamics and memory.

## Results

**Chromatin information reflects TF-chromatin interaction patterns.** We first aimed to quantify patterns of chromatin accessibility around TF-chromatin interactions. We reasoned that TF binding creates a localized impact on chromatin architecture, which may result in TF-specific signatures. To measure chromatin architecture, we focused on the assay for transposase-accessible chromatin using sequencing (ATAC-seq)[7], that can simultaneously quantify both TF and nucleosome signatures, which are reflected in the ATAC-seq fragment length patterns. This chromatin architecture can be visualized using V-plots[8], which show the aggregate ATAC-seq fragment midpoints around

TF binding sites and can result in a stereotyped "V" pattern of points for bound TFs with well-phased adjacent nucleosomes (Fig. 1a, upper plot). The extent of organization in the V-plot can be measured using Shannon's entropy equations[5] to quantify information. We, therefore, calculated the information content of the ATAC-seq fragment size distribution around TF binding sites as a way to quantify V-plot organization (Fig. 1a, middle plot). To adjust for potential bias arising from non-uniform ATAC-seq fragment coverage across the V-plot, we devised a metric called chromatin information enrichment (CIE; Methods, Fig. 1a, middle and lower plots, Supplementary Fig. 1). We summarized CIE into a single value, named feature V-Plot Information Content Enrichment (f-VICE), which represents the CIE at landmark TF and nucleosomal positions across the V-plot. These positions are expected to have high CIE when the nucleosomes are well-positioned around the TF binding site (Fig. 1a, lower plot). Therefore, f-VICE quantifies the degree of chromatin architecture organization around a TF.

We initially focused on the GM12878 lymphoblastoid cell line, for which there is high-quality, deeply-sequenced ATAC-seq data[7] and 41 TF chromatin immunoprecipitation followed by sequencing (ChIP-seq) experiments that pass our inclusion criteria (Supplementary Data 1)[9]. To increase our ability to detect TF-chromatin interactions, we generated an independent GM12878 ATAC-seq dataset with higher signal-to-noise ratio (measured by TSS enrichment and fraction of reads in peaks; Supplementary Fig. 2). Using these datasets, we created V-plots and calculated f-VICEs centered on bound motif instances for 41 TFs. The ATAC-seq fragment pattern was most ordered around CTCF, a known chromatin organizer[10], where we detected clusters of fragments distributed periodically in a "V" pattern indicating nucleosome phasing (Fig. 1b-c, Supplementary Fig. 3). CTCF f-VICE was highest among the 41 TFs (Fig. 1d). Other TFs, exemplified by AP-1 and NFKB, had diverse f-VICEs (Fig. 1b-d, Supplementary Fig. 3). The TF f-VICE values were highly

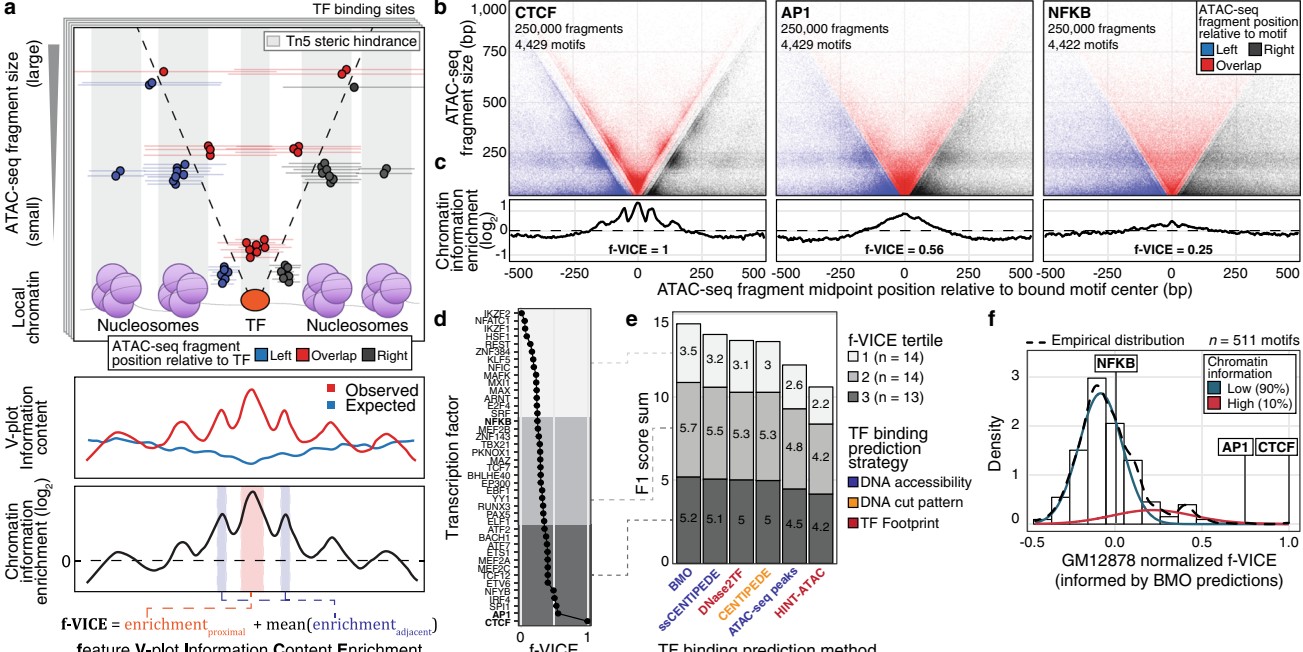

**Fig. 1 Information content of TF-chromatin interactions. a** Upper: TF binding impacts the chromatin architecture and the observed ATAC-seq fragment distribution around TF binding sites. Middle and bottom: calculation of CIE and f-VICE. **b**, **c** V-plots and CIEs of CTCF, AP-1, and NFKB (GM12878 ATAC-seq data generated in this study). V-plots were downsampled to highlight differences in chromatin architecture (but not for f-VICE calculation). **d** f-VICEs calculated for TFs with GM12878 ChIP-seq data. **e** F1 score sum of TF binding prediction algorithms. **f** Normalized GM12878 BMO-informed f-VICE distribution.

concordant across GM12878 ATAC-seq libraries with distinct signal-to-noise metrics (Spearman's ρ range 0.81–0.98, median = 0.96, $n = 7$ libraries; Supplementary Fig. 4a, b) and downsampled datasets representing sequencing depths as low as 5 million high-quality alignments (Spearman's ρ range 0.89–0.99, median = 0.98; Supplementary Fig. 4c–e). We additionally calculated f-VICE values separately for motif instances across quintiles of ChIP-seq signal intensity, chromatin accessibility, and TF affinity. We found that the f-VICE values were highly correlated across the motif subsets (Spearman's ρ range 0.70–0.89, median = 0.85; Supplementary Fig. 5a-b). We performed a linear regression of f-VICE controlling for these potentially confounding metrics and found that the model coefficients for the TF terms were highly correlated with their respective f-VICEs (Spearman's ρ = 0.9, $p < 1e-323$; Supplementary Fig. 5c), which demonstrates the robustness of the f-VICE metric. These results indicate extensive differences in TF-chromatin interactions, which are captured in the CIE patterns.

**Footprint-free prediction of TF binding and chromatin information.** One alternative to determine f-VICEs for TFs without ChIP-seq data is to rely on binding predictions using chromatin accessibility data. This motivated us to first evaluate the performance of current TF binding prediction algorithms. Most algorithms search for footprints, which are regions of low chromatin accessibility embedded within larger accessible regions, thought to be caused by cleavage protection from bound TFs[11–13]. However, a recent report indicated that ~80% of TFs do not associate with footprints[14]. Hence, we developed BMO ("Bee MOdel of TF binding"), an unsupervised method to predict TF binding using negative binomial models of chromatin accessibility[15–17] and co-occurring motifs[18], without relying on footprints (Supplementary Fig. 6, Methods). We benchmarked BMO and other TF binding prediction methods (DNase2TF[12], HINT-ATAC[13], PIQ[11], and CENTIPEDE[19]) using TF ChIP-seq data from GM12878 and HepG2 ($n = 41$ and $n = 59$, respectively; Supplementary Data 1). DNase2TF, HINT-ATAC, and PIQ rely on footprints to predict TF binding, while CENTIPEDE learns informative DNA cut patterns indicating TF binding. To compare methods, we calculated their F1 scores for predicting each TF. We additionally developed a custom implementation of CENTIPEDE that does not rely on the DNA cut patterns around the motif, which we named signal-sum CENTIPEDE (ssCENTIPEDE; Methods). ssCENTIPEDE allows us to estimate the contribution of the DNA cut patterns around the TF motif compared to motif accessibility in CENTIPEDE predictions (Supplementary Note). BMO had overall higher performance (higher F1 score than the other methods in 74% of all comparisons; Supplementary Fig. 7). Importantly, the footprint-agnostic methods (BMO, CENTIPEDE, and ssCENTIPEDE) outperformed the footprint-based methods (higher F1 scores in 90% of comparisons), particularly on TFs with low f-VICEs (Fig. 1e, Supplementary Figs. 7-11; Supplementary Note). To determine if the overall lower performance of the footprinting-based methods resulted from only sites with strong TF affinity being associated with footprints, we benchmarked all the methods separately for TF binding sites in the top and bottom 20% of TF occupancy, based on ChIP-seq signal (Supplementary Figure 12). The footprinting-based method DNase2TF had higher performance compared to BMO when predicting the top 20% occupancy binding sites (Supplementary Figure 12a). However, the increase in performance for DNase2TF was limited to medium and high f-VICE TFs and only when using the high signal-to-noise GM12878 ATAC-seq dataset generated for this study (Supplementary Fig. 12a, left). BMO outperformed DNase2TF in the top 20% binding sites when using

the Buenrostro et al. GM12878 dataset[7] (Supplementary Fig. 12a, right), which had a lower signal-to-noise ratio (Supplementary Fig. 2). These results are consistent with (1) only a subset of high-occupancy binding sites from high f-VICE TFs associating with detectable footprints and (2) footprint detection being sensitive to sample quality. Together, these findings indicate that footprinting-based approaches are not the optimal strategy to predict TF binding. Instead, we show that TF binding is generally more accurately predicted using a simple chromatin accessibility model tuned to each TF motif.

**Chromatin information varies across TFs.** Having determined that our BMO footprint-agnostic method is among the most accurate for predicting TF binding regardless of their f-VICE, we proceeded with BMO predictions to estimate f-VICEs for TFs without ChIP-seq data. BMO-predicted f-VICEs were significantly correlated with f-VICEs calculated from TF ChIP-seq data across all datasets (Pearson's ρ range 0.72–0.79, median = 0.76, all $p < 9.78e-11$; Supplementary Fig. 13). We therefore concluded that BMO can be used to estimate f-VICEs without ChIP-seq data and performed BMO TF binding predictions to calculate f-VICEs for 540 non-redundant TF motifs (Supplementary Data 2-3). We used high-quality ATAC-seq datasets from four additional human tissues (pancreatic islets[20], pancreatic islet sorted alpha and beta cells[21], and CD4 + cells[22]; Supplementary Data 1), selected by applying a strategy that uses the highly stereotyped chromatin architecture in ubiquitous and conserved CTCF/cohesin binding sites to measure sample quality (Supplementary Fig. 14) (Methods). We normalized f-VICEs within each sample to control for differences in the number of bound motif predictions and overall chromatin accessibility (Supplementary Fig. 15). Among the 540 motifs, we observed a mixture of two f-VICE distributions and therefore used a mixture of two Gaussians to fit the data. The median percentage of high f-VICE motifs across datasets was 14% (range 7–18%, Fig. 1f, Supplementary Fig. 16), which is comparable to the percentage of motifs associated with DNase footprint protection across datasets (median = 19%) from another study[14] and supports our conclusion that footprint-based algorithms will not perform well on the majority of TFs (median of 86% across datasets). Together, these results reinforce the use of footprint-agnostic methods like BMO for accurately calculating f-VICE. Importantly, our results suggest that only a subset of TFs associate with highly organized chromatin.

**Chromatin information is associated with TF-DNA residence times.** TF residence time, which corresponds to the duration of DNA binding for a TF, is an important biophysical measurement that can influence TF activity[2,23]. Based on the high f-VICEs for CTCF and AP-1 and low f-VICE for NFKB (Fig. 1c, d), which agree with the known residence times for these TFs (Supplementary Table 1), we hypothesized that CIE correlates with residence time. We correlated BMO-informed f-VICEs with previously measured fluorescence recovery after photobleaching (FRAP) data from mammalian cell lines (Supplementary Table 1), which provide an upper bound of TF residence time[24,25]. Using a robust linear regression to protect against outlier influence, we found that f-VICE was significantly associated with FRAP recovery times in all samples (β range 0.7–1.3, median = 0.98, all Bonferroni-adjusted $p \leq 0.001$; Fig. 2a, Supplementary Fig. 17). This suggests that TFs associated with high CIE have longer residence times.

A recent study found that cohesin has a residence time 10- to 20-fold higher than CTCF[25]. We reasoned this difference could be reflected in the local chromatin architecture and calculated the

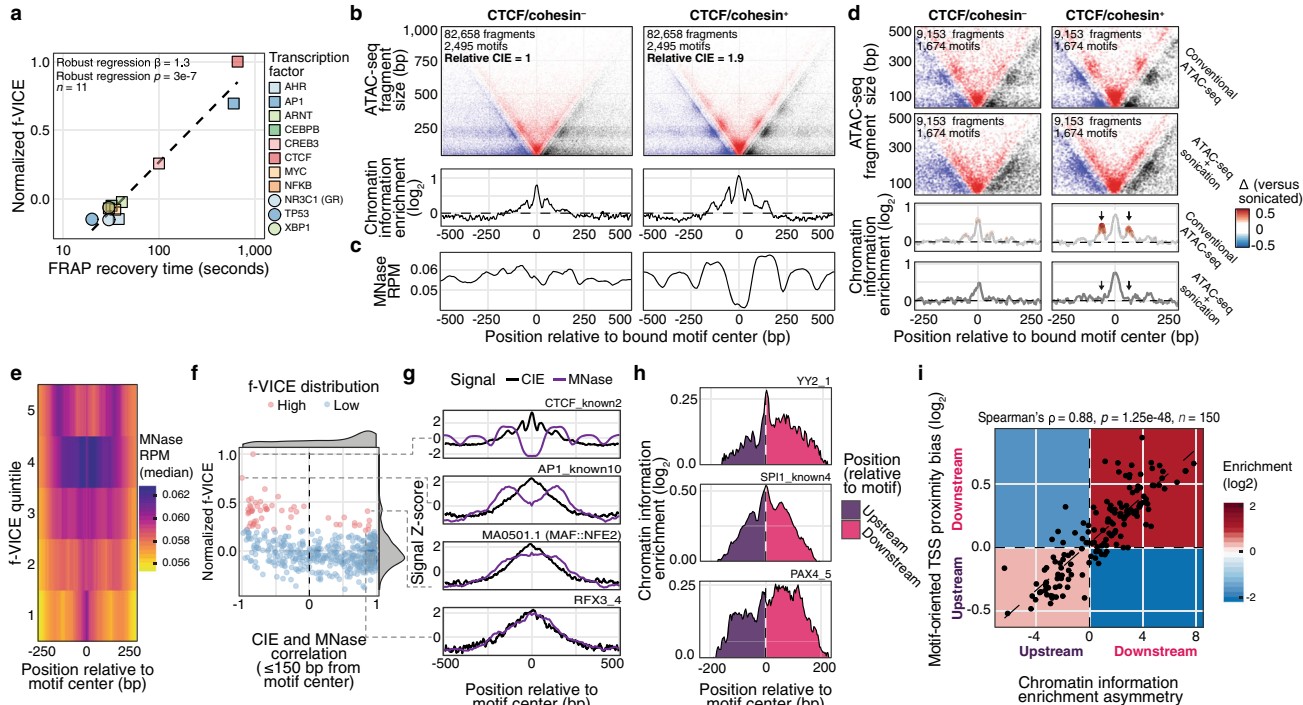

**Fig. 2 Chromatin information informs residence times and TF-nucleosome interactions. a** Correlation of FRAP recovery times and GM12878 f-VICEs. Dashed line, linear model fit. **b** V-plots and CIEs of CTCF/cohesin⁺ and CTCF/cohesin⁻ motifs. **c**, GM19238 MNase-seq reads per million mapped reads at the same motifs. **d** CTCF/cohesin⁺ and CTCF/cohesin⁻ motifs in the sonicated and conventional GM12878 ATAC-seq data. Colors, differences relative to sonicated. **e** Median nucleosome occupancy across motifs from different f-VICE quintiles in the GM12878 ATAC-seq data generated in this study. **f** Scatter plot of CIE and MNase correlation ≤ 150 bp from the motif as function of f-VICE. **g** High f-VICE motifs associated with nucleosome phasing (top) or a well-positioned nucleosome at the motif region (bottom). **h** Top 3 motifs with CIE asymmetry Z-scores in GM12878. **i** Scatter plot of motif-oriented TSS position bias and CIE asymmetry in TSS-proximal motifs. Enrichments calculated by permuting the signs of observed values ($n = 10{,}000$).

CIE of the GM12878 lymphoblastoid cell line CTCF binding sites with and without the presence of cohesin (CTCF/cohesin⁺ and CTCF/cohesin⁻), controlling for ATAC-seq coverage, ChIP-seq signal, and motif quality (Supplementary Fig. 18a). CTCF/cohesin⁺ had 1.9-fold higher CIE compared to CTCF/cohesin⁻ (Fig. 2b, Supplementary Fig. 18b), indicating these distinct CTCF occupancy classes have different CIE signatures. We next compared the nucleosome positioning signals inferred from lymphoblastoid cell line micrococcal nuclease sequencing (MNase-seq) profiles (Supplementary Data 1). Only the CTCF/cohesin⁺ class had phased nucleosomes around the binding site (Fig. 2c, Supplementary Fig. 18c), consistent with longer residence times associated with nucleosome phasing. To experimentally validate these results, we generated chromatin accessibility data using a modified ATAC-seq protocol with an additional sonication step (Methods) to disrupt the fragment size information (Supplementary Fig. 19). There were no detectable nucleosome phasing patterns in the motif-flanking CIE signature (~50 bp from the motif) of the sonicated sample (vertical arrows in Fig. 2d and Supplementary Fig. 19b), which we determined was not due to size-selection bias in the library preparation (Supplementary Fig. 19c). These results are complementary to our residence time results above in that they show our CIE approach can capture differences in chromatin organization in subsets of TF binding sites that are associated with different residence times.

**Most high chromatin information TFs associate with nucleosome phasing.** To systematically characterize the association between CIE and nucleosome positioning, we compared GM12878 CIE patterns across TF motifs to the nucleosome positions obtained both from ATAC-seq using the NucleoATAC algorithm[26] and from lymphoblastoid MNase-seq profiles (Supplementary Fig. 20). High f-VICE motifs had lower nucleosome occupancy directly at the motif region and phased nucleosomes directly adjacent to it (Fig. 2e, Supplementary Fig. 20e). Accordingly, the CIE patterns of high f-VICE motifs were significantly more likely to be anti-correlated with the MNase-seq signal at the motif region ($p = 2.18\mathrm{e}{-}13$, generalized linear model; Fig. 2f,g, upper two panels, Supplementary Fig. 20f). We calculated the degree of nucleosome phasing around the motif region and found that it was significantly correlated with f-VICE (Pearson's $\rho = 0.4$, $p = 3.60\mathrm{e}{-}22$; Supplementary Fig. 20g). However, we observed that 22% of the high f-VICE motifs had high MNase signal at the motif region (12/54; Fig. 2g, bottom two panels, Supplementary Fig. 21). This indicates that high CIE patterns more commonly result from nucleosome phasing induced by TF binding, but can also result from a well-positioned nucleosome at the TF binding site. The latter case is consistent with TFs that bind at the nucleosome dyad[4,27] and include a member of the RFX family[4]. These results indicate that the CIE levels reflect the overall level of chromatin organization and can capture different nucleosomal configurations. Therefore, the CIE patterns are more general and complementary to nucleosome positioning data.

**Chromatin information asymmetry at TF motifs.** Previous reports suggested that a subset of TFs directionally bind DNA, with potential effects on gene regulation[11,28,29]. To investigate this further, we extended our information content analyses to quantify CIE asymmetry (Methods). Of the 540 motifs tested, 150 had significantly asymmetric CIE (Bonferroni corrected $p < 0.05$;

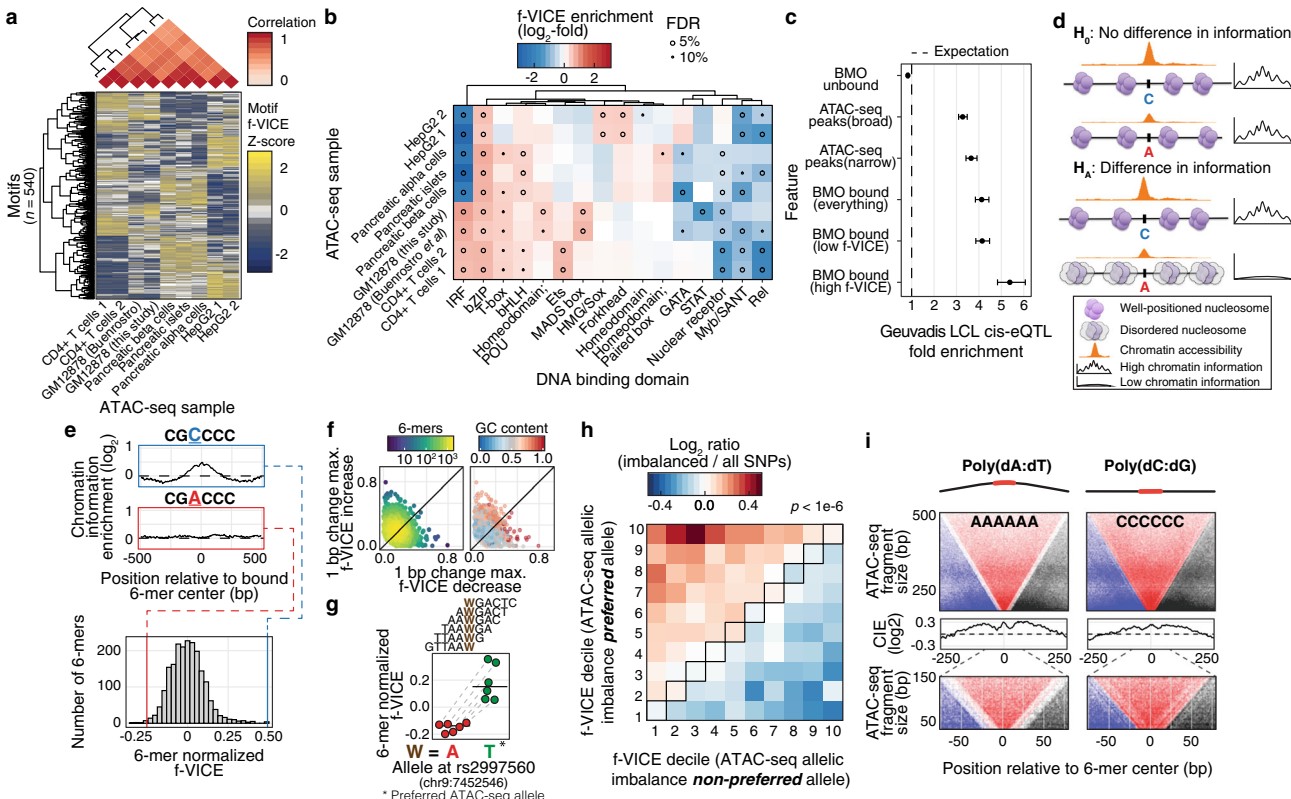

**Fig. 3 The chromatin information landscape of human tissues. a** Hierarchical clustering of f-VICE Z-scores. **b** f-VICE enrichments across DBDs. **c**, LCL *cis*-eQTLs enrichments (*n* = 2,743 lead SNPs). Central dot, effect size mean; error bars, effect size SD. **d** Hypothesis schematic. **e** Upper: Two 6-mers with 1-bp difference in sequence. Lower: pancreatic islets 6-mer normalized f-VICE distribution. **f** Range of f-VICE differences associated with 1-bp difference in 6-mer sequence. **g** Predicted f-VICE change associated with rs2997560 in pancreatic islets. Horizontal bars, median. **h** Log₂ ratio of f-VICE decile changes associated with the preferred and non-preferred alleles of imbalanced SNPs versus all tested SNPs in pancreatic islets. **i** CIE patterns at DNA 6-mers with poly(dA:dT) and poly(dC:dG) sequences.

Fig. 2h, Supplementary Fig. 22a). The direction of CIE asymmetry was significantly correlated with the direction of the nearest TSS relative to each motif instance (Spearman's $\rho = 0.66$, $p = 3.34e$-20; Supplementary Fig. 22b). To determine if asymmetric CIE was an artifact of TSS proximity, we calculated CIE asymmetry separately for TSS-proximal ($\leq 1$ kb) and TSS-distal ($\geq 10$ kb) motif instances. The TSS-distal and TSS-proximal CIE asymmetry directions agreed significantly more than expected by chance (111/150, binomial test $p = 3.38e$-9, Fisher exact test $p = 2.80e$-5; Supplementary Fig. 22c-d), suggesting that CIE asymmetry is intrinsic to the TF motif. The magnitude of asymmetry was higher in TSS-proximal motifs (Supplementary Fig. 22d), suggesting that TSS proximity amplifies TF CIE asymmetry. Accordingly, the correlation between nearest TSS direction and CIE asymmetry was stronger at TSS-proximal motifs (Spearman's $\rho = 0.88$, $p = 1.25e$-48; Fig. 2i). These results indicate that a subset of TFs are associated with asymmetric TF-chromatin interactions.

**Chromatin information patterns are tissue-specific and associate with genetic control of gene expression**. We next aimed to investigate cross-tissue differences in CIEs. We performed an unsupervised hierarchical clustering of motif f-VICEs and found that it recapitulated the expected tissue grouping (Fig. 3a). A recent study demonstrated that NF-KB (p65) residence time is determined by its DNA-binding domain (DBD)[30], which motivated us to ask if DBDs are associated with CIE. We assigned DBDs and protein domains to motifs and designed a permutation-based rank test to calculate domain f-VICE enrichments (Methods). We observed both common and tissue-

specific f-VICE enrichments, including IRF and ETS in blood-related samples, PAX in islet-related samples, and HMG/SOX and FOX domains in HepG2 (FDR < 10%; Fig. 3b, Supplementary Fig. 23). Our findings show the landscape of TF-chromatin interactions varies across tissues and reflects protein domain-level TF properties.

The prevalence of tissue-specific differences in CIEs led us to examine the role of high f-VICE TFs in regulating gene expression. We calculated the enrichment of the motifs categorized as high or low f-VICE in GM12878 (Fig. 1f) to overlap lymphoblastoid *cis*-expression quantitative trait loci (*cis*-eQTLs) datasets[31,32], which represent gene expression genetic control regions. High f-VICE motifs had 15–30% higher (median = 24%) fold-enrichment in *cis*-eQTLs compared to low f-VICE motifs (Fig. 3c, Supplementary Fig. 24a), but no differences in eQTL effect sizes (Supplementary Fig. 24b). These results indicate that high f-VICE TFs are more likely to mediate genetic effects on gene expression, but not their magnitude.

**High chromatin information TF motifs are associated with increased chromatin accessibility**. Given that high f-VICE TFs have highly ordered chromatin (Fig. 1), high predicted residence times (Fig. 2a, b, Supplementary Fig. 17), and nucleosome phasing properties (Fig. 2e, Supplementary Fig. 21), we hypothesized that their regulatory effects (Fig. 3c) could result from acting as or recruiting pioneer factors that induce chromatin accessibility[11,33]. Supporting this, we find that motifs belonging to families associated with known pioneers[11,33,34] had some of the highest f-VICEs in all (e.g. OCT/POU, KLF) or in a subset of

samples (e.g. FOXA in HEPG2, PAX in islets, IRF and BATF in GM12878 and CD4 + cells; Fig. 3b, Supplementary Fig. 21, and Supplementary Data 3). If true, we would expect increased CIE at single nucleotide polymorphism (SNP) alleles associated with increased chromatin accessibility (i.e. with ATAC-seq allelic imbalance; Fig. 3d). We performed a motif-agnostic approach to calculate the f-VICEs associated with every DNA 6-mer, controlling for differences in chromatin accessibility. This strategy allows the interrogation of genetic variants by determining all the possible DNA 6-mers formed by each allele and their corresponding f-VICEs, without incurring in bias from underrepresented sequences in the TF motif library. DNA 6-mers have a distribution of f-VICEs (Fig. 3e; Supplementary Fig. 25a), and GC-pure 6-mers had the highest f-VICEs (Supplementary Fig. 25b), which is consistent with GC-rich sequences driving enhancer activity[35] and suggests that high GC-content regions represent anchors of nuclear architecture. Notably, a single basepair change can lead to large differences in 6-mer f-VICEs (Fig. 3e-f, Supplementary Fig. 25c-e), suggesting that genetic variation impacts CIE. To test this, we determined f-VICEs for 6-mers formed by both alleles at SNPs with significant ATAC-seq allelic imbalance (binomial test $p < 0.05$) in GM12878 and pancreatic islets (Methods). The preferred ATAC-seq alleles were significantly biased to form higher f-VICE 6-mers compared to the less favored allele in all samples (all $p < 2.81e-4$, permutation tests; Fig. 3g-h, Supplementary Fig. 26). These findings support a model where TFs with potential pioneer-like properties bookmark regions of the genome to allow binding of other migrant-like TFs[11,33]. Accordingly, TF motifs that are predictive of binding without any chromatin accessibility data (based solely on the motif match score) have significantly higher f-VICEs in GM12878 and HepG2 (robust linear regression $p \leq 0.001$; Supplementary Fig. 27). This suggests that high f-VICE TFs, particularly CTCF, are more likely to bind any strong motif regardless of its underlying accessibility, while the remaining TFs require motifs located in already accessible regions.

**"Non-canonical" chromatin information shape patterns reflect local DNA topology.** The f-VICE metric we developed here summarizes the CIE pattern into a single value, and it was tuned to detect patterns associated with TF binding, which have higher information signal proximal to the TF motif (Fig. 1a, Supplementary Fig. 1). However, other genomic features may have distinct CIE patterns that would not be captured by the f-VICE equation. This motivated us to investigate if there is a subset of features associated with "non-canonical" V-plot patterns. We found a subset of DNA 6-mers that had higher CIE levels at regions distal to the 6-mer center (Supplementary Fig. 28a-c). These DNA 6-mers were characterized by stretches of three or more consecutive A or T nucleotides. This is consistent with these 6-mers overlapping A-tracts, which are regions characterized by poly(dA:dT) nucleotides and are associated with curvature in the DNA molecule and nucleosome organization[36]. The CIE pattern observed in the poly(dA:dT) 6-mers likely results from low Tn5 integration at the 6-mer region, which was not observed in poly(dC:dG) 6-mers (Fig. 3i). In addition, we observed that the poly(dA:dT) 6-mers were generally associated with a phased nucleosome pattern in the MNase-seq data (Supplementary Fig. 28e), consistent with this DNA topology favoring nucleosome exclusion[36]. Of note, the MNase patterns in the poly(dA:dT) 6-mers were almost indistinguishable from to the ones associated with nucleosome phasing at high f-VICE motifs (Fig. 2g, upper panels, Supplementary Fig. 21), illustrating how the CIE metric is complementary to nucleosome positioning data. We additionally analyzed a subset of DNA 11-mers ($n = 1,248$) predicted to have

regulatory properties in GM12878 (Methods) to determine if any of these features were associated with less common CIE patterns. We did not find evidence that these 11-mers had CIE patterns differing from the TF motifs (Supplementary Fig. 28f-h), indicating that sequences associated with regulatory activity have higher proximal CIE levels. Together, these results demonstrate how local DNA topology is associated with higher-order chromatin organization reflected in the CIE patterns.

**Asymmetrical CIE patterns at transcription start sites.** While we focused this study on TF-chromatin interactions, the CIE framework presented here can be used to study other genomic features. To illustrate this, we generated V-plots and calculated CIE for the TSS regions from highly expressed genes in GM12878. Using this approach, we can observe the highly asymmetrical accessibility pattern in the TSS, indicating a well-positioned +1 nucleosome downstream of the TSS (Supplementary Fig. 29). This demonstrates the versatility of our entropy-based methodology to characterize genomic features which would otherwise require laborious experimental approaches or would not be possible in vivo.

## Discussion
In this study, we develop and use entropy-based algorithms to analyze chromatin accessibility data and quantify the level of chromatin organization at genomic features of interest. This chromatin information approach is more general and complementary to analyzing nucleosome positioning, as it captures additional features such as DNA protection from TF binding and local DNA topology. We use this entropy-based approach to dissect TF-chromatin interaction patterns across human cell lines and tissues. The TF-chromatin interactions are captured in the information content patterns of chromatin accessibility and reflect functional properties of TFs, such as TF-DNA residence times, specific protein domains, and the ability to induce nucleosome repositioning. We find that a subset of TFs (10–20%) have high chromatin information and are more highly associated with the genetic control of both chromatin accessibility and gene expression. We hypothesize that these TFs define cell state by potentially acting as pioneers. Future studies are necessary to experimentally determine the fraction of TFs associated with high chromatin information that have pioneer properties.

A potential application of the methodology presented here is to estimate chromatin information patterns in other organisms, including non-model organisms where less information about their TF repertoire is available. We reason that the unbiased estimation of information patterns encoded in DNA substrings occurring in accessible chromatin, such as we performed using DNA 6-mers (Fig. 3e), can be used to inform the possible chromatin organization configurations associated with that organism. Such an approach could potentially be used to determine the appearance of pioneer-like TFs along the eukaryotic tree, which would be reflected in the emergence of a long right tail in the f-VICE distribution (Fig. 3e and Supplementary Fig. 25a).

One limitation of our methodology is that it can be affected by clusters of TFs binding in close proximity, which can potentially decrease the apparent information of the local chromatin. This can be circumvented with careful experimental approaches to separate these TF binding sites, such as the one we used for CTCF and cohesin. In addition, the f-VICE metric we developed here is highly tuned to detect patterns associated with TF binding, but other genomic features may have distinct CIE patterns. Indeed, we found a subset of DNA 6-mers with "non-canonical" V-plot patterns, which we hypothesized reflected a specific DNA topology (Fig. 3i, Supplementary Fig. 28). Therefore, it is reasonable to

expect the existence of less common genomic features with "non-canonical" CIE patterns. One approach to systematically detect these features is through unsupervised clustering of the CIE shape patterns.

Finally, we show that footprinting-based algorithms to predict TF binding, which remain a popular choice in the field, are sensitive to the TF-chromatin information landscape we describe. We develop and cross-validate a tool for predicting TF binding based on chromatin accessibility that outperforms footprinting-based methods. These findings represent strong evidence that most TF binding sites do not associate with footprints. Collectively, our results show a dynamic landscape of TF-chromatin interactions, with implications for gene regulation and cell state memory.

## Methods

**GM12878 cell culture**. GM12878 cells were obtained from the Coriell Institute for Medical Research. We cultured cells following the ENCODE GM12878 cell culture protocol (www.encodeproject.org/documents/1bb75b62-ac29-4368-9855-68d410e1963a). Cells were cultured at 37 °C (5% $CO_2$) with added plasmocin (Invivogen, San Diego, CA; 50 μg/mL) to the RPMI 1640 (2 mM L-glutamine, 15% fetal bovine serum) growth media to prevent mycoplasma contamination.

**GM12878 ATAC-seq data generation**. We conducted ATAC-seq following the protocol described in Ref. [37] with two modifications: (1) we used as input 250,000 cells and (2) we used a home-made Tn5 enzyme[38] (described in the Tn5 Synthesis section below). After harvesting and centrifuging GM12878 cells (5 min × 500 g at 4 °C), we incubated a suspension of 250,000 cells with 12.5 μL of 1:1 mix of Tn5 enzyme that carry 5-methylC-MEDS-A oligos and MEDS-B oligos at 37 °C for 30 min in a 50 μL reaction. We column-purified the tagmented DNA using the Zymo DNA Clean & Concentrator-5 kit (Zymo Research, Irvine, CA) and constructed Illumina sequencing library using the Kitzman lab custom indexing primers (barcode plates #5 and #10; Supplementary Table 2). We PCR-amplified a total of 11 cycles until the amplification curve reached its mid-log phase (1/3 to 1/2 of max signal), and then purified the PCR products using SPRI beads prepared using a homemade mix[39] by combining Carboxyl-modified Sera-Mag Magnetic Speed-beads (Fisher Scientific, cat. #65152105050250) in a PEG/NaCl buffer (0.1% beads, 18% PEG-8000 w/v, 1 M NaCl, 10 mM Tris-HCl, pH 8.0, 1 mM EDTA, pH 8.0) and eluted in 22 μL of TTE8 buffer. Sequencing was performed on an Illumina HiSeq 4000 platform at the University of Michigan Sequencing Core and a total of ~33 million paired-end 52 bp reads were generated.

**Sonicated GM12878 ATAC-seq data generation**. For each replicate we incubated 250,000 cells with three different concentrations of enzyme (0.2X, 1X, and 5X; 1X corresponds to 2.5 μL of Tn5 that carry 5-methylC-MEDS-A oligos) at 37 °C for 30 min in a 50 μL reaction. We column-purified the tagmented DNA using the Zymo DNA Clean & Concentrator-5 kit (Zymo Research, Irvine, CA), and sonicated to ~350 bp using the Covaris M220 sonicator (peak incident power - 50 W; duty factor - 20%; cycles per burst - 200; treatment time - 60 s). We constructed Illumina sequencing library using the ACCEL-NGS Methyl-seq DNA Library kit (Swift Biosciences #DL-ILMMS-12; revision 160106) with the following modifications to the manufacturer's protocol: (1) We skipped "Ligation" and "Post-ligation SPRI" steps (pg. 10-11), as the 5' end of the fragments had already been tagged during the transposition step. Accordingly, we eluted DNA with 20 μL of TTE8 (10 mM Tris-HCl, 0.1 mM EDTA, 0.05% Tween-20, pH 8) for Post-Extension SPRI step (pg. 10), instead of 15 μL, to adjust for the difference in volume before proceeding to the "Indexing PCR" step (pg. 11); (2) We used a 2:1 beads:sample ratio for "Post-Extension SPRI" step (pg. 10) and 1.8:1 beads:sample ratio for "Post-PCR SPRI" step (pg. 12); and 3) For indexing PCR, we used the Kitzman lab custom primers (barcode plate #5; Supplementary Table 2) to prime the P5 end and the "IndexD7XX" primers (Swift Biosciences #38096, previously #DI-ILMMS-48; Supplementary Table 2) to prime the P7 end. We PCR-amplified a total of 14 cycles for 0.2X, 1X samples and 16 cycles for 5X samples until amplification curve reached its mid-log phase (1/3 to 1/2 of max signal), and then purified the PCR products using SPRI beads prepared as in using a homemade[39] mix by combining Carboxyl-modified Sera-Mag Magnetic Speed-beads (Fisher Scientific, cat. #65152105050250) in a PEG/NaCl buffer (0.1% beads, 18% PEG-8000 w/v, 1 M NaCl, 10 mM Tris-HCl, pH 8.0, 1 mM EDTA, pH 8.0) and eluted in 22 μL of TTE8 buffer. Sequencing was performed on an Illumina HiSeq 2500 platform at the University of Michigan Sequencing Core and a total of ~33 million paired-end 126 bp reads were generated.

**Tn5 synthesis**. Tn5 synthesis was performed according to a previously described protocol[38]. We used the pTXB1-Tn5 plasmid vector (Addgene #60240, generously gifted by Dr. Rickard Sandberg) to synthesize the hyperactive Tn5 allele with E54K and L732P mutations but wild-type at M56 and transformed into T7

Express *lysY/I$^q$* Competent *E. coli* (NEB cat#3013H). We grew in 1 L of LB culture until it reached A600 = 0.9, added IPTG to a final concentration of 0.25 mM, and then incubated for an additional 4 h at 23 °C before harvesting by centrifugation (20 min at 5,000 rpm) and freezing overnight at −70 °C. After thawing the cell pellet on the next day, we resuspended it in HEGX buffer (20 mM HEPES-KOH pH 7.2, 0.8 M NaCl, 1 mM EDTA, 10% glycerol, 0.2% Triton X-100) containing 1X complete protease inhibitors (Roche; 1 tablet per 50 mL) and lysed them in an ice-cold metal beaker using a metal-tip sonicator at the Center for Structural Biology core at the University of Michigan (10 cycles of 30 bursts, 50% duty cycle, output 6; output power ~20–40 W). The lysate was pelleted by centrifugation (15,000 RPM for 30 min at 4 °C in the Beckman JA17 rotor). 2.1 mL 10% neutralized PEI (Sigma P3143) was added to the supernatant dropwise on a magnetic stirrer. The precipitate was removed by centrifugation at 12,000 rpm for 10 min at 4 °C (JA17 rotor). We assembled the Tn5 transposome using the on-column transposase assembly method[38]. First, we transferred the supernatant to a Kontes Flex protein purification column that contained 8.33 mL of chitin resin, and then incubated on a nutator for 1.5 h at 4 °C. We drained the column, and washed the column four times with 25 mL of HEGX buffer for each wash (i.e. a total of 100 mL wash). The washed resin was transferred to 50 mL conical tubes and mixed with the Tn5-MEDS-A or Tn5-MEDS-B annealed oligos (for each mL of resin, we mixed in 200 μL of MEDS duplex oligos at 250 μM). The mixture was incubated at room temperature on a nutator. After 24 h, we washed the resin three times with HEGX buffer to remove unbound MEDS oligos. After the third wash, we resuspended the resin in HEGX buffer containing 100 mM DTT, and incubated at 4 °C on a nutator for 41 h. After incubation, we poured the resin solution to a Kontes flex column, and then drained it to collect eluates (which contained the released Tn5-MEDS complexes) in a 50 mL conical tube. We transferred the collected eluates to Snake-Skin Dialysis Tubing (Thermo #68100; 10,000 MWCO; 22 mm × 35 feet dry diameter) and dialyzed against two changes of 2X Tn5 dialysis buffer (100 mM HEPES-KOH pH 7.2, 0.2 M NaCl, 0.2 mM EDTA, 2 mM DTT, 0.2% Triton X-100, 20% glycerol) at 4 °C for 24 h. We measured the protein concentration using a Bradford assay (Bio-Rad protein assay) and adjusted the concentration to 12.5 μM with 50% glycerol and 1X dialysis buffer.

**ATAC-seq data processing**. Reads were trimmed for barcodes using cta (v. 0.1.2) and aligned to the hg19 reference human genome using BWA mem (v. 0.7.15)[40] similarly to our previous study[41], with additional parameters -I 200,200,5000 to avoid larger ATAC-seq fragments being discarded. We removed duplicate alignments using Picard (broadinstitute.github.io/picard; v2.8.1) and retained properly paired and uniquely mapped alignments with high mapping quality using samtools view (v. 1.3.1)[42] with flags -f 3 -F 4 -F 8 -F 256 -F 1024 -F 2048 -q 30. We called broad and narrow peaks using MACS2 (v. 2.1.1.20160309)[43] with flags -g hs -nomodel -shift -100 -extsize 200 -B [-broad] -keep-dup all and kept peaks that did not intersect blacklisted regions by the ENCODE consortium due to poor mappability (sites.google.com/site/anshulkundaje/projects/blacklists), using bedtools (v2.26.0)[44], and that reached 5% FDR. All data was processed uniformly using Snakemake[45].

**Motif processing**. We used the PWM scans from[41]. Briefly, we used biallelic SNPs and short indels from the 1,000 Genomes project (release v5)[46] to generate comprehensive scans with FIMO (v0.5.4)[47], using the background nucleotide frequencies from hg19 and a $p < 1e-4$. We only kept motif instances that intersected mappable regions and did not intersect blacklisted regions. In order to reduce motif redundancy, we performed PWM clustering in our motif database using the *matrix-clustering* tool from RSAT (v1.0.5)[48], with parameters -lth cor 0.7 -lth Ncor 0.7. For each of the 540 clusters obtained, we used the motif with the highest total PWM information content for downstream analyses.

**V-plots, chromatin information enrichments, and f-VICEs**. V-plots[8] were generated by determining the size and position of all ATAC-seq fragments within ±500 bp of the genomic feature set of interest (e.g. bound TF motifs). The fragments size and positions were obtained using the script *measure_signal*, which is part of atactk, a suite of tools to analyze ATAC-seq data developed for this study (https://github.com/ParkerLab/atactk). We excluded from the V-plots any instances of the genomic feature of interest that were closer than 500 bp to each other to avoid interference, and we also excluded ATAC-seq fragments smaller than 40 bp. We generated a matrix encoding the fragment counts per position relative to the feature (−500 to 500 bp) and per fragment size. To decrease sparsity, we summed the fragment counts corresponding to each fragment size across consecutive positions using a sliding window of 10 bp width in 8-bp increments between windows (e.g. [−500,−490], [−492,−482], …). We did not sum fragment counts across fragment sizes. We then calculated the normalized information content ($I(x)$) of the vector of fragment size counts for each 10-bp window using Eq. (1), where $H(x)$ is the maximum-likelihood Shannon's entropy function implemented of the entropy R package (v. 1.2.1)[49], and $H_{max}$ is the maximum Shannon's entropy

for a vector of the same length (i.e. the maximum fragment size minus 40).

$$I(x) = 1 - \frac{H(x)}{H_{max}} \quad (1)$$

To additionally control for sparsity from low ATAC-seq coverage, we calculated the expected normalized information content by repeating all the steps described above using a randomized version of the input ATAC-seq fragment list. The randomized data from each genomic feature was obtained by randomly permuting the labels corresponding to the fragment sizes and positions. The null expectation allowed us to calculate the chromatin information enrichment (CIE), defined as the $\log_2$ of the observed normalized information content divided by the expected normalized information content. Therefore, the CIE metric allows for a more direct comparison between distinct genomic features compared to information content alone because it controls for the differences in coverage between features.

**Pseudocode**. normalized_info = []
 for position i in (i = −500, i ≤ 490, i += 8):
 start = i; end = i + 10
 fragment_sum = []
 for fragment_size j in (j = 41, j = max (fragment_size), j += 1):
 fragment_sum[j] = sum j[k] for k in (k = start, k = end, k += 1)
 normalized_info[i] = normalized_information (fragment_sum)
 expected_information = permute fragment size labels & re-calculate normalized_info
 CIE = $\log_2$ (normalized_info / expected_information)

To summarize the CIE vector of a given genomic feature, we designed a metric called feature V-plot information content enrichment (f-VICE). We calculated f-VICE using Eq. (2).

$$f - VICE = \sum_{i=-25}^{i=25} CIE_i + \frac{\sum_{i=-70}^{i=-50} CIE_i + \sum_{i=50}^{i=70} CIE_i}{2} : \quad (2)$$
$$(CIE_i > 1 \rightarrow CIE_i) \wedge (CIE_i \leq 0 \rightarrow 0) \wedge (i \in \mathbb{Z})$$

The [−25,25] and [−70,−50 50,70] coordinates correspond to the TF-proximal and TF-adjacent CIE peaks in the CTCF V-plot, respectively. These landmark positions are expected to have high CIE values when the TF associates with nucleosome phasing, due to the abundance of small fragments spanning the TF binding site (proximal peak) and positioned between the TF and the first pair of flanking nucleosomes (adjacent peaks; Fig. 1a and Supplementary Fig. 1). To further control for differences in the number of motif instances and accessibility between the motifs analyzed in this study, the f-VICE values were normalized across all motifs using the residuals of the linear model $f\text{-}VICE \sim \log_{10}(m) + \log_{10}(f)$, where $m$ corresponds to the number of motif instances and $f$ corresponds to the total number of ATAC-seq fragments at the predicted bound motif instances for each motif (Supplementary Fig. 15). The residuals for each sample were then divided by the corresponding CTCF value in the corresponding sample to normalize it to 1. For simplicity, we refer to the CTCF-normalized residuals as "normalized f-VICE" and the CTCF-normalized f-VICEs (without linear regression) as "CTCF-normalized f-VICE" throughout this study. The linear model normalization step was not performed in any of the ChIP-seq f-VICE analyses due to lack of data points to accurately fit the linear model. In Fig. 1c, the CTCF, AP-1, and NFKB V-plots were downsampled to equal number of ATAC-seq fragments and motifs by selecting the top $k$ motifs ranked by the number of ATAC-seq fragments within ±500 bp, and then further downsampling to 250,000 fragments, where $k$ represents the number of bound NFKB motifs (the lowest among the three TFs in Fig. 1c). These downsampled V-plots were only used for visual comparison and not used for CIE and f-VICE calculations.

For the analyses in Supplementary Fig. 5, we divided the TF-bound motifs into quintiles of ChIP-seq signal (*signalValue* column of the ENCODE narrowPeak file), ATAC-seq signal (number of ATAC-seq fragments ±100 bp from the bound motif center), and TF affinity (obtained using Eq. (4) of[50], which is a proxy of the strength of the TF-DNA interaction[51]). To avoid the TF affinity values being affected by the haplotype-aware PWM scans in GM12878, we only used bound TF motif instances detected in the reference genome. We then calculated f-VICEs using the subset TF binding sites belonging to each metric quintile and compared the corresponding f-VICE values to the f-VICE values obtained from all binding sites. To perform the linear regression analysis in Supplementary Fig. 5c, we used the rank-based inverse normal transformed f-VICE values as dependent variable to a model that included indicator variables to each TF, to the corresponding signal type (ChIP, ATAC, TF affinity) from which the quintiles were calculated, and an ordinal variable corresponding to the quintile.

**Additional ATAC-seq samples selection**. In addition to the GM12878 ATAC-seq dataset generated for this study, we analyzed eight additional publicly available datasets corresponding to pancreatic islets[20,21], CD4+ cells[22], GM12878[7], and HepG2[52]. With the exception of HepG2, these datasets were selected from a survey of all the public ATAC-seq datasets available until the end of 2017. We selected for our analyses datasets with at least 20 million high-quality autosomal reads and transcription start site (TSS) enrichment ≥ 6. In addition, we only retained samples with the stereotypical chromatin information enrichment indicative of nucleosome phasing at ubiquitous and conserved CTCF-cohesin binding sites. These CTCF

binding sites provide a reference V-plot with expected high accessibility and periodical chromatin information enrichment patterns in any high-quality sample. The ubiquitous and conserved CTCF-cohesin sites were defined as CTCF motifs overlapping ENCODE CTCF and Rad21 ChIP-seq peaks in at least in at least 54/59 (CTCF) and two (Rad21) different human tissues, located in bi-directionally mappable regions between human and mouse using bnMapper[53] that also corresponded to CTCF motif matches in the mm9 reference genome. To quantify samples, we defined our high-quality GM12878 dataset as a reference and calculated the CIE correlation ≤ 200 bp from the motif center. Samples with Spearman's ρ < 0.8 were discarded (Supplementary Fig. 14). Finally, we only retained tissues/cells that had at least two samples that passed our stringent selection criteria. A list of all dataset accessions used in this study can be found in Supplementary Data 1.

**BMO transcription factor binding prediction**. BMO ("Bee" MOdel of TF binding) builds on previous reports that the degree of chromatin accessibility around a motif[15–17] and the presence of co-occurring motifs[18] positively correlate with TF occupancy, corresponding to the analogy of TFs behaving as "Brownian bees" in the genome looking for TF binding sites ("flowers"). The more accessible and the greater quantity of flowers, the more likely the bee will interact with them. BMO uses per-TF negative binomial models of the motif accessibility and density signals to estimate the likelihood of a TF motif instance being bound. BMO performs three steps: (1) calculate the background ATAC-seq fragment negative binomial distribution, (2) calculate the co-occurring motifs negative binomial distribution, and (3) combine the $p$ values from the two distributions (Supplementary Fig. 6).

Using all genomic matches for a given motif PWM, BMO calculates the number of ATAC-seq fragments overlapping a region ±100 bp from every motif instance, ignoring fragments that integrate directly in the motif coordinates. The latter step is performed to mitigate ATAC-seq bias, as the nucleotide sequence in the motif regions is relatively constant across features and is more subject to assay-specific bias compared to the motif-flanking regions. BMO uses the ATAC-seq fragment counts ±100 bp from motif instances occurring outside ATAC-seq peaks to fit a negative binomial distribution which corresponds to the ATAC-seq background for that motif. For computational reasons, BMO randomly selects 10,000 motif instances outside peaks to fit the ATAC-seq background, repeats this step 100 times, and uses the average mean and overdispersion parameters for the ATAC-seq background. This approach is 1-2 orders of magnitude faster compared to fitting the ATAC-seq background negative binomial distribution on the entire set of motif matches outside ATAC-seq peaks and yields identical results. BMO then calculates the $p$ values for the number of ATAC-seq fragments ±100 bp from every motif match based on the ATAC-seq background distribution for that motif. Next, BMO determines the number of additional instances of the same motif PWM within ±100 bp of every motif instance. This is used to fit a second negative binomial representing the distribution of co-occurring motifs within ±100 bp of each motif instance.

BMO combines the nominal $p$ values of the ATAC-seq and co-occurring motifs distributions by summing their Z scores[54]. This step yields a single $p$ value representing chromatin accessibility and number of co-occurring motifs. A given motif instance will have more significant $p$ values if it is located in accessible chromatin and/or have many instances of the same motif nearby. Multiple testing correction was performed using the Benjamini-Yekutieli correction procedure[55]. Motif instances are reported as bound when the adjusted $p$ value < 0.05. Fitted NB distributions were obtained using the R packages MASS (v. 7.3-50)[56] and fitdistrplus (v. 1.0-11)[57].

**CENTIPEDE**. For each motif, we generated a strand-specific (relative to the motif orientation) base-pair resolution matrix encoding the number of Tn5 transposase integration events in a region ±100 bp from each motif instance using *make_cut_matrix* with parameters *-d -r 100*. The cut matrix and the vector of motif PWM scores were used as input for CENTIPEDE (v. 1.2)[19]. Any motif instance was considered bound if the CENTIPEDE posterior probability was higher than 0.99. The *make_cut_matrix* script was developed as part of atactk (https://github.com/ParkerLab/atactk).

**Signal-sum CENTIPEDE (ssCENTIPEDE)**. For this study, we developed an alternative implementation of CENTIPEDE, called signal-sum CENTIPEDE (ssCENTIPEDE). ssCENTIPEDE differs from CENTIPEDE in that it is blinded to any positional patterns encoded in the Tn5 cut preferences. To run ssCENTIPEDE, we performed CENTIPEDE predictions using as input the vector of motif PWM scores and a vector encoding the sum of Tn5 DNA cuts in the vicinity of each motif instance (instead of a base-pair resolution matrix encoding the positions of the DNA cuts relative to the motif). This strategy informs the overall motif instance accessibility while omitting any positional patterns that can be used by CENTIPEDE to predict TF binding.

**DNase2TF**. In order to run DNase2TF (v. 1.0)[12] on ATAC-seq data, we offset all the cut points calculated using *paired_end_bam2split.r* by 4 bp before using them as input to the software, which was run with default parameters. We intersected the outputted footprints coordinates with each motif bed file and considered bound any motif instance that intersected a footprint scored with FDR < 0.05.

**HINT-ATAC**. We performed footprinting analyses with HINT-ATAC (RGT v. 1.1.1)[13] using as input the broad ATAC-seq peaks and filtered BAM file from each sample. In their methods, the authors used MACS2 narrow peaks, but we found that they had lower performance compared to broad peaks (Supplementary Fig. 8), so we used the latter for the analyses. We intersected the HINT output file with each motif file and considered bound any motif instance that intersected a footprint.

**PIQ**. We performed PWM scans using the *pwmmatch.exact.r* script included with PIQ (v. 1.3)[11]. BAM files were processed with *bam2rdata.r* due to an error in the code of *pairedbam2rdata.r* which prevented any of our BAM files from being processed. Footprinting was performed using the *pertf.r* script. Because PIQ performs its own PWM scans, we compared PIQ to BMO only on PWM matches that were shared between PIQ and BMO (using bedtools intersect).

**Dataset downsampling**. In order to compare f-VICE calculations or TF binding prediction methods across multiple sequencing depths, we uniformly downsampled BAM files using the *-s* flags of samtools view (v1.9), which downsamples files while maintaining read pairs intact (this behavior is not present in version 1.3.1). These downsampled files were used as input to generate V-plots or for peak calling and all other steps required prior to running each TF binding prediction method.

**TF binding evaluation**. We defined as true positives for a given TF all motif matches that fully intersected a ChIP-seq ENCODE conservative irreproducible discovery rate (IDR) narrow peak in the respective sample (using the flag *-f 1.0* in bedtools intersect). We only analyzed TFs that had motifs in our database and at least 1,000 bound motif instances. For TFs with multiple PWMs, we selected the PWM with the highest total information. For TFs with multiple ChIP-seq experiments, we selected the one with the highest number of bound motifs. To evaluate methods, we calculated the area under the precision-recall curve (AUC-PR), which informs the performance of the classifier in ranking bound and unbound motif instances, and the F1 score, which takes into account threshold used to call bound motif instances. We did not use areas under the receiver-operator characteristic curve (AUC-ROC) given the highly skewed class imbalance between bound and unbound motifs, which makes AUC-ROCs an unreliable metric to evaluate TF binding predictions[58,59]. AUC-PRs were calculated using packages ROCR (v. 1.0-7) and PRROC (v. 1.3) in R[60,61]. To rank predictions, we used the -log₁₀ adjusted $p$ values for BMO, the number of reported tags from HINT-ATAC, the posteriors calculated by CENTIPEDE and ssCENTIPEDE, the -log₁₀ $p$ values calculated by DNase2TF, the purity score outputted by PIQ, and MACS2 -log₁₀ $p$ values for motifs in peaks. F1 scores were calculated using Eq. (3) at the following thresholds for each method: BMO adjusted $p$ value < 0.05, CENTIPEDE and ssCENTIPEDE posterior ≥ 0.99, any motif instance overlapping a HINT-ATAC predicted footprint, any motif instance overlapping a DNase2TF predicted footprint with FDR value < 0.05, and any motif instance predicted bound by PIQ.

$$F1 = 2 \cdot \frac{\text{precision} \cdot \text{recall}}{\text{precision} + \text{recall}} \qquad (3)$$

For the analyses in Supplementary Fig. 12, we used the *signalValue* column of the ENCODE narrowPeak files to divide the data into quintiles. We used either the top or bottom quintiles for benchmarking the TF binding prediction methods, after removing from the analyses any motif instances that intersected the remaining quintiles.

**Mixture models for f-VICE distributions**. High and low f-VICE Gaussian mixture model distributions were fitted using the R package mixtools (v. 1.1.0)[62] using as input the normalized f-VICEs for each ATAC-seq sample, after filtering low signal motifs where the number of total predicted bound instances for the motif was in the lowest decile of that sample. The filtering step is to avoid potential bias for lower f-VICE values due to sparsity. We used a posterior probability of 0.5 as the threshold to split the high and low f-VICE distributions. We alternatively tried to fit a single Gaussian distribution to the f-VICEs instead of a mixture model. The Bayesian information criterion values of the Gaussian mixture models were greater than the single Gaussian models in all nine samples analyzed in this study, indicating that a mixture model was a better fit for these data. Q-Q plots show that the Gaussian mixture model and single Gaussian model thresholds can be used to obtain a similar separation between the low and high f-VICE motifs (Supplementary Fig. 16b), which indicates the robust nature of the chosen thresholds.

**FRAP/f-VICE robust regression and CTCF-Cohesin regions comparisons**. To measure the correlation between FRAP recovery times and f-VICE, we performed a literature search of reported FRAP recovery times, which are referenced in Supplementary Table 1. Robust linear regressions of f-VICE and FRAP recovery times were performed with the rlm function of the R package MASS (v. 7.3-50)[56]. For each TF with FRAP recovery times, we used the f-VICE from the motif with highest total PWM information content in our database. f-VICEs for these motifs were normalized using the same linear regression model described earlier, but

including all the motifs in our database ($n = 1,850$). For each sample, we required that the gene corresponding to each TF had RNA-seq TPM ≥ 1 in a related tissue in GTEx (except for pancreatic islets, where we used the RNA-seq data from[20]).

CTCF-Cohesin regions in GM12878 were obtained by selecting CTCF motifs that intersected conservative IDR GM12878 CTCF ChIP-seq peaks (ENCODE accessions ENCFF096AKZ, ENCFF710VEH, and ENCFF963PJY) and the merged GM1287 RAD21 optimal IDR peaks (ENCODE accessions ENCFF753RGL and ENCFF002CPK). CTCF regions without cohesin were obtained similarly as above, but removing CTCF motifs that intersected any of the GM12878 RAD21 ChIP-seq peaks. All operations were performed with bedtools (v. 2.26.0). The choice of optimal IDR peaks for RAD21 aimed to increase the number of RAD21 peaks are included in the CTCF-cohesin⁺ regions, therefore increasing the stringency of the comparisons. We performed a quantile-based downsampling approach to make the CTCF/cohesin⁺ and CTCF/cohesin⁻ regions comparable regarding ChIP-seq signal, ATAC-seq signal, and FIMO motif scores. This was done by selecting all CTCF motifs encompassing the CTCF/cohesin⁺ and CTCF/cohesin⁻ regions and, for each feature (ATAC-seq fragments, ChIP-seq signal, or motif scores), calculating quantiles ($n = 20$). Then, for every quantile, we counted the number of motifs belonging to the CTCF/cohesin⁺ and CTCF/cohesin⁻ regions and randomly downsampled the group with more motifs instances to have the same number of motifs as the other in that quantile. This ensured that both regions had the same number of motifs and comparable distributions of ATAC and ChIP signals and motif scores (as an example of this normalization, refer to Supplementary Fig. 18a).

**Pseudocode**. for feature in {ATAC, ChIP, PWM}:
 split feature in 20 quantiles
 for quantile in {1..20}:
 set1 = CTCF/cohesin⁺ ∈ feature$_{\text{quantile}}$
 set2 = CTCF/cohesin⁻ ∈ feature$_{\text{quantile}}$
 smallest_set = smallest (set1, set2)
 largest_set = largest (set1, set2)
 n = size (smallest_set)
 randomly select n items from largest_set

For the main figures, we used CTCF and RAD21 experiments ENCFF963PJY and ENCFF753CPK, respectively (the same comparisons using the other CTCF/RAD21 datasets are presented in Supplementary Fig. 18). The quantity labeled as relative chromatin information enrichment in Fig. 2b corresponds to the sum of positive chromatin information enrichment (above dashed line) in each V-plot, divided by the CTCF/cohesin⁻ value for normalization.

**Clustering**. Cross-tissue clustering and dendrograms (Fig. 3a) were calculated using the Euclidean distances of the pairwise Spearman correlation of f-VICEs across samples. Normalized f-VICE values were converted to motif-wise Z scores before clustering.

**Nucleosome occupancy analyses**. We used NucleoATAC (v. 0.3.4)[26] to estimate the nucleosome positions in Fig. 2e. Briefly, we ran the software with default parameters using as input the GM12878 ATAC-seq data generated in this study. We calculated the aggregate density of nucleosome midpoints positions relative to the BMO predicted bound motif coordinates for each motif (see Supplementary Fig. 20g for an example density plot). We then converted the nucleosome density values into Z-scores, binned the motifs into f-VICE quintiles, and calculated the average nucleosome density Z-score per f-VICE quintile (Supplementary Fig. 20h).

Paired-end MNase unmapped reads from the lymphoblastoid cell line GM19238 were obtained from SRA, under accession SRR452483[63]. Reads were mapped to the hg19 reference using BWA mem and processed in an identical fashion to the ATAC-seq data, with an additional step to retain only sequenced fragments of length $147 \pm 2$ bp, therefore enriching for mononucleosomal fragments. The MNase aggregate signal plots were generated using ngsplot (github.com/shenlab-sinai/ngsplot; v. 2.63). For each motif plot, we used as input the BED files corresponding to the regions that were used to generate the corresponding V-plot. Motif MNase Z-scores were calculated using the MNase reads per million mapped reads (RPM) signal tracks outputted by ngsplot and Eq. (4). MNase/CIE correlations were calculated using positions ≤150 bp from the motif center.

$$Z(x) = \frac{x - \text{mean}(X)}{sd(X)} \qquad (4)$$

**Chromatin information enrichment asymmetry**. Chromatin information enrichment asymmetry was calculated as the log₂ ratio between the positive information content enrichment in the left and right of the motif center. To estimate significance, we used a permutation test where each fragment midpoint had a 50% chance of changing its direction relative to the motif while keeping the same distance (*i.e.* multiply its x-axis value by −1). We calculated the asymmetry of the permuted V-plots ($n = 100,000$) to generate a null distribution of asymmetry. Because the null was normally distributed based on Kolmogorov-Smirnof and Shapiro normality tests, we were able to estimate $p$ values beyond the number of permutations by calculating the observed asymmetry Z-score relative to the null

distribution. To calculate the nearest TSS directionality bias, we counted the number of active protein-coding TSS (GENCODE V19) (determined with the presence of LCL Cap analysis gene expression (CAGE) tag clusters, described in the next session) on the left and right sides of the motif and calculated the $\log_2$ ratio of the two sides. For the proximal and distal motif V-plots, we restricted our analyses to motifs occurring ≤1 kb or ≥10 kb from the nearest CAGE-supported TSS of any type (e.g. lincRNAs, pseudogenes; GENCODE V19). Enrichments of the plots in Fig. 2f were calculated by randomly permuting the signal of the points in the x- and y-axis (n = 10,000 permutations).

**CAGE tag cluster identification**. We downloaded CAGE data (fastq files) for 154 LCL samples[64] and mapped to hg19 using STAR (version 2.5.4b; default parameters)[65] and pruned the mapped reads to high quality reads (using samtools view v. 1.3.1; options -F 4 -q 255). We used the paralu method[66] to identify clusters of CAGE start sites (CAGE tag clusters). We called TCs in each individual sample using raw tag counts, requiring at least 2 tags at each included start site and allowing single base-pair tag clusters ('singletons') if supported by >2 tags. We then merged the tag clusters on each strand across samples. For each resulting segment, we calculated the number of LCL samples in which TCs overlapped the segment. We included the segment in the consensus TCs set if it was supported by independent TCs in at least 10 individual LCL samples, resulting in n = 10 tag clusters. We then filtered out regions blacklisted by the ENCODE consortium due to poor mappability using bedtools (v. 2.26.0) to obtain the final set of LCL tag cluster regions.

**DNA binding domain enrichments**. DNA binding domains (DBD) enrichments were performed using a f-VICE rank sum permutation test. We assigned DBDs to the non-redundant motifs that mapped between our database and the one reported in[67], which has manually curated DBD-motif assignments. In order to map motifs between databases, we used tomtom[68] and selected motif matches with p-value < 0.05 after a conservative Bonferroni adjustment using all comparisons as denominator (i.e. number of motifs in our database times the number of motifs in the queried database), which yielded high-confidence DBD assignments for 402 of 540 motifs. We used the f-VICE rank from each motif to calculate the f-VICE rank sum the DBD and compared the observed value to a null distribution of 100,000 rank sums obtained from randomly permuting gene labels. This approach ensures that all the DBD retain their sizes during each permutation. We retained DBDs with at least 5 motifs and calculated the f-VICE enrichments for each DBD using the $\log_2$ of observed f-VICE rank sum divided by the median of the null. FDR was calculated separately per sample, using the empirical p-value from the 100,000 permutations. We simultaneously performed a similar analysis using InterPro protein domains (v. 72)[69] (Supplementary Fig. 23). In order to assign domains to motifs, we first mapped our motifs to CIS-BP database (Build 1.02)[70], which has high-confidence motif-gene assignments, and retained genes that mapped to a single motif using the same approach described above. Each gene was then linked to a motif f-VICE score (n = 475) and we only retained domains with at least 5 genes after motif-gene mapping. Permutation and enrichments were calculated identically as described above.

***cis*-eQTL enrichments**. Feature enrichments in eQTLs were calculated using GREGOR (v. 1.2.1)[71] and QTL tools fenrich (v. 1.1)[72]. We used the lymphoblastoid cell line (LCL) eQTLs sets from Geuvadis[32] and GTEX[31] (FDR < 5%). GREGOR background estimations were performed using SNPs with LD 0.99 for eQTL, with a maximum distance of 1 Mb from the variants of interest. Variants used as input for GREGOR were pruned to have maximum linkage disequilibrium $r^2$ of 0.8 with any other variant. For fenrich, we used the most significant SNP per gene as input.

**ATAC-seq allelic imbalance analyses**. To determine SNP allelic bias in ATAC-seq data, we used the publicly available data from Buenrostro et al., listed in Supplementary Data 1, the Parker Lab GM12878 sample discussed here, and the ABCU196 islet sample[20]. For GM12878 data, adapters were trimmed using cta (v. 0.1.2), and reads mapped to hg19 using bwa mem (default options except for the -M flag). Bam files were filtered to high-quality autosomal read pairs using samtools view (v. 1.3.1) with flags -f 3 -F 4 -F 8 -F 256 -F 2048 -q 30. WASP (v. 0.2.1, commit 5a52185; python version 2.7)[73] was used to diminish reference bias; for remapping the reads as part of the WASP pipeline, we used the same mapping and filtering parameters described above for the initial mapping and filtering. Duplicates were removed using WASP's rmdup_pe.py script. We used the phased GM12878 VCF file downloaded from ftp://ftp-trace.ncbi.nlm.nih.gov/giab/ftp/release/NA12878_HG001/NISTv3.3.1/GRCh37/HG001_GRCh37_GIAB_high conf_CG-IllFB-IllGATKHC-Ion-10X-SOLID_CHROM1-X_v.3.3.1_highconf_phased.vcf.gz. To avoid double-counting alleles, overlapping read pairs were clipped using bamUtil clipOverlap (v. 1.0.14; http://genome.sph.umich.edu/wiki/BamUtil:_clipOverlap). For the Buenrostro et al. data, the bam files from the samples in Supplementary Data 1 were then merged to create a single GM12878 bam file using samtools merge (v. 1.3.1). For each heterozygous autosomal SNP, we then counted the number of reads containing each allele, using only bases with base quality of at least 20. We used a two-tailed binomial test that accounted for reference allele bias to evaluate the significance of the allelic bias at each SNP (as

described in[20]; when calculating the expected fracRef, SNPs in the top 25th percentile of read coverage were downsampled to the 50th percentile coverage and SNPs with coverage less than 10 were excluded). When performing the binomial test, we downsampled the coverage at each SNP such that each SNP had coverage = 20 (to reduce coverage-related biases). The islet ATAC-seq data was processed and tested as described in[20], except that we also downsampled coverage at each SNP to 20 reads when performing the binomial test. We did not test SNPs in regions blacklisted by the ENCODE Consortium because of poor mappability (wgEncodeDacMapabilityConsensusExcludable.bed and wgEncodeDukeMapabilityRegionsExcludable.bed). We retained for downstream analyses all loci with nominally significant binomial p values (p < 0.05) and at least 2 reads (10%) mapped to any allele. After selecting loci that intersected at least one predicted bound 6-mer (see next section), we obtained 257, 1,449, and 15,802 imbalanced SNPs for the two GM12878 datasets (this study and Buenrostro) and pancreatic islets, respectively.

**6-mer f-VICE calculations**. We generated a list of all possible DNA 6-mers (n = 2016 after filtering reverse complements) and scanned the hg19 reference genome to obtain the coordinates for all their corresponding matches. Similarly to motifs, we only retained 6-mer matches that were in mappable regions and did not intersect blacklisted regions. For each set of 6-mer matches, we used BMO to determine the subset that was predicted bound. We calculated the normalized f-VICE for each 6-mer using exactly the same steps as in the motifs, including using linear regression to control for chromatin accessibility. For each 6-mer, we determined its immediate neighbors in sequence space (every 6-mer that differed by exactly 1 letter; Hamming distance = 1) and calculated the f-VICE differences between each of the neighbors relative to the original 6-mer. We then calculated the Euclidean distance between the neighbors with highest and lowest f-VICE to determine what was the f-VICE range associated with that 6-mer.

For each locus with significant allelic imbalance, we calculated the f-VICE associated with all six 6-mer instances overlapping with each allele. For each sample, we determined the f-VICE decile changes associated with every SNP tested for allelic imbalance and determined the matrix of the $\log_2$ ratios of each possible decile change in the imbalanced versus all tested SNPs. To test for significance, we devised a permutation test where all symmetrical pairs of f-VICE deciles, $x_{ij}$ and $x_{ji}$ for $i,j \in \{1, 2, \ldots, 10\}$, had a 50% chance of switching their $\log_2$-ratio values in each permutation (n = 1,000,000 permutations).

**TSS V-plots**. To generate the GM12878 TSS V-plots in Supplementary Fig. 28, we used all the protein coding genes (GENCODE V19) in the highest quintile of gene expression based on GTEx data (EBV-transformed lymphocytes). We only selected TSS instances that were more than 500 bp away from each other.

**LS-GKM**. The V-plots and CIE patterns in Supplementary Fig. 29e-f were obtained the top-scoring 11-mers from LS-GKM (commit 164a4a4)[74] calculated using the ATAC-seq peaks from the GM12878 sample generated in this study. We analyzed all 11-mers with scores ≥ 1 (n = 1,248) based on the inflection of the score distribution of all 11-mers with positive scores (Supplementary Fig. 29f). Because of the low representation of 11-mers in the genome, which precluded the use of BMO to predict bound 11-mers, we instead generated V-plots from all the 11-mer instances overlapping the GM12878 ATAC-seq peaks used as input for LS-GKM. CIE Z-score clusters were obtained using the R k-means implementation, using parameters k = 5 and 1,000 random starts.

**Reporting summary**. Further information on research design is available in the Nature Research Reporting Summary linked to this article.

## Data availability

GM12878 ATAC-seq data is available at GEO under accession "GSE135074". All processed data are available at https://doi.org/10.5281/zenodo.3478583. All other relevant data supporting the key findings of this study are available within the article and its Supplementary Information files or from the corresponding author upon reasonable request.

Publicly available datasets used in this study (detailed in Supplementary Data 1):

ATAC-seq datasets: SRX1497362, SRX1497365, SRX2717887, SRX2717888, E-MTAB-7543 (rep 3), E-MTAB-7543 (rep 1), SRX298000, SRX2768920, SRX2768919, SRX2768918, SRX2768917, SRX298001, SRX298002, SRX298003, SRX298004, SRX298005, and SRX298006.

MNase-seq datasets: SRR452483

ChIP-seq datasets (GM12878): ENCFF784PEF, ENCFF794KET, ENCFF133GHG, ENCFF969FVF, ENCFF748WOQ, ENCFF006MIL, ENCFF096AKZ, ENCFF850MAC, ENCFF382VEJ, ENCFF880NTF, ENCFF476RII, ENCFF565SXH, ENCFF742XOI, ENCFF662JYS, ENCFF343VAG, ENCFF337XDI, ENCFF708VKT, ENCFF762AZG, ENCFF112CKJ, ENCFF407JNK, ENCFF288RYL, ENCFF811FYS, ENCFF006MAM, ENCFF138CXP, ENCFF861YUL, ENCFF359EFT, ENCFF269LZJ, ENCFF739VBA, ENCFF363BLT, ENCFF969EMZ, ENCFF618KHI, ENCFF936XYD, ENCFF147DQK, ENCFF040ZUY, ENCFF069KRU, ENCFF515HWO, ENCFF759PVA, ENCFF817AOQ, ENCFF294BZJ, ENCFF369JYP, and ENCFF229WZB.

ChIP-seq datasets (HepG2): ENCFF841GUR, ENCFF215TZZ, ENCFF510UKU, ENCFF614HYY, ENCFF502XEV, ENCFF467TLM, ENCFF419CGD, ENCFF996YOX, ENCFF329MOF, ENCFF786XDZ, ENCFF066MGX, ENCFF811CMJ, ENCFF392RQB, ENCFF653JVV, ENCFF816PYF, ENCFF860ZQP, ENCFF105NJB, ENCFF722GFS, ENCFF888ZAZ, ENCFF081YHC, ENCFF682HTY, ENCFF421ZII, ENCFF044GIB, ENCFF222YCT, ENCFF267YOX, ENCFF916SBW, ENCFF596XCU, ENCFF025GBN, ENCFF645NAR, ENCFF814UDQ, ENCFF762QHA, ENCFF754RXP, ENCFF634BXB, ENCFF266HXY, ENCFF249FDQ, ENCFF773PSQ, ENCFF487OSZ, ENCFF920ITW, ENCFF489SNI, ENCFF180BOV, ENCFF618IBW, ENCFF076BOS, ENCFF859UHG, ENCFF645MNT, ENCFF907PFI, ENCFF003NDR, ENCFF315DYP, ENCFF942TIQ, ENCFF164KYP, ENCFF810LYT, ENCFF138XKG, ENCFF263VIF, ENCFF568BTO, ENCFF834ACA, ENCFF016NAS, ENCFF962DEO, ENCFF858QVH, ENCFF363GGS, and ENCFF004UHW.

A reporting summary for this Article is available as a Supplementary Information file. Source data are provided with this paper.

## Code availability

Code and scripts used for the analyses performed in this study are publicly available at http://github.com/ParkerLab/chromatin_information[75]. BMO and atactk are publically available at http://github.com/ParkerLab/BMO[76] and http://github.com/ParkerLab/atactk[77].

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

## Acknowledgements

We thank members of the Parker Lab, L.J. Scott, P. Freddolino, P. Wittkopp, M. Burmeister, G. Higgins, P. Pereira, M. Puthenveedu, and J. Brancho for helpful comments. Dr. Rickard Sandberg generously gifted the plasmid used for Tn5 synthesis. Sequencing was performed at the UM Sequencing Core Facility. This work was supported by the ADA Pathway to Stop Diabetes Grant 1-14-INI-07 and by the National Institute of Diabetes and Digestive and Kidney Diseases grant R01 DK117960 to S.C.J.P.

## Author contributions

R.O.A.: Analyzed data, designed computational experiments, wrote the manuscript. Y.K.: Generated ATAC-seq datasets. J.H.: Implemented computational algorithms. A.V.: analyzed eQTL and chromHMM data. P.O. calculated ATAC-seq allelic imbalance. J.K. designed ATAC-seq experiments. S.C.J.P.: designed experiments, analyzed data, wrote the manuscript, and supervised all aspects of the project.

## Competing interests

All authors declare no competing interests.
