## [Peer Review File · Nature Communications]

Reviewers' comments:

Reviewer #1 (Remarks to the Author):

The paper "Chromatin information content landscapes inform transcription factor and DNA interactions" by D'Oliveira Albanus et al. describes a novel approach to analyze and understand transcription factor binding properties by the analysis of ATAC-seq signals. The paper builds on a framework borrowed from information theory and makes large usage of a particular type of analysis (V-plots), previously conceived by the group of W. Greenleaf, to analyze global patterning of nucleosome dyads around transposase accessible sites.

Given the novelty of the approach, I would expect better characterization of the proposed metrics. Calculation of CIE and f-VICE as reported in methods remains a bit obscure, I would suggest some schematic representation (and pseudocode) to better render the underlying idea (although it can be glimpsed by Fig1A lower panel).

In the methods section it is stated that fragments were downsampled to produce figures (Fig1b), although original values were used to perform all the analysis. This prompts me a simple question that should be addressed when CIE and f-VICE are introduced: how these values are affected by coverage in the experiment? Indeed the coverage of ATAC-seq experiments varies from 20-30M to 200M fragments, depending on the analysis scope. Since high coverage is suggested to perform footprint analysis, is this true also for information analysis? What would be the minimal and the optimal coverages for this kind of analysis? Some hints of this dependence are clear in Supplementary Figure 10 (and, in fact, depth is used as covariate to calculate normalized f-VICE).

CIE is summarized into a single value, f-VICE. The authors should discuss if the function is bijective or not and, in case it is surjective, which V-plot configurations could lead to high f-VICE values. I expect "well organized" patterns (as CTCF) always lead to high f-VICE values, but it would be interesting to know if any other configuration, possibly more disorganized, may lead to similar values. I imagine this could be tackled theoretically and by simulating different V-plot configurations.

Related to the previous two questions: how does noise influence CIE and f-VICE? There may be ATAC-seq experiments with some background noise, is the approach described robust to this? Also, could one use the framework presented to estimate the level of noise in ATAC-seq analysis?

BMO is introduced in this paper, some benchmarks were presented and discussed in the supplementary note. I would try to briefly summarize them in the main text.

The bimodal distribution of normalized f-VICE scores is, in my opinion, loosely captured by gaussian models. In supplementary figure 11 and, notably, in figure 1f, the modal value of the high-valued scores is poorly recapitulated by the gaussian distribution represented by the red line. I wonder what data have been used to build the gaussian model (possibly the data in supplementary figure 10B?). Maybe a simple Q-Q plots on the distributions displayed could be used to choose cutoffs for the two mixtures.

There is a little discrepancy between the protocol for sonicated ATAC-seq (targeting to 350bp fragment size) and the curve in Supplementary Figure 14A, where the size distribution peaks at ~200 bp. Also, it is not clear to me if the purification step before sonication selects smaller DNA fragments (<2000 bp) or if all chromatin is eluted, although this won't probably affect results.

When testing TSS-distal/proximal CIE asymmetry a binomial test is used. While 111/150 is an

impressive fraction and I'm persuaded it is a good result, a better test would be a fisher test for distal/proximal and agree/disagree, especially if the intended scope of the test is "To determine if asymmetric CIE was an artifact of TSS proximity".

Lastly, I have a more general comment on the methods presented in this paper: once the list of genomic intervals for ATAC sites is computed, they can be iteratively sampled to build V-plots and calculate CIE/f-VICE. The resulting scores will distribute in a multimodal way, as expected from the results presented. If the sampling strategy is opportunely tuned, this procedure should tend to a distribution of scores representative of all TF that could bind chromatin (in theory) or at least to some major clusters (e.g. low-CIE, mid-CIE, high-CIE). Do the authors think such approach could be useful and applied to study chromatin properties in non-model organisms? If yes, I suggest to add a comment in the discussion.

Minor comments:

- if BMO is an acronym, it is never explained in the text
- page 7, fig 1g is mentioned but the panel is missing from the figure and its legend
- I suggest a general revision of methods to improve clarity.

Davide Cittaro

Reviewer #2 (Remarks to the Author):

In the manuscript entitled "Chromatin information content landscapes inform transcription factor and DNA interactions" Albanus et al. devise a metric for chromatin organization at specific positions of the genome that they term "chromatin information content", and use it to classify different transcription factor ChIP-seq peaks, motif matches and 6 bp subsequences. They find that features such as higher residency time of transcription factor and genetic effects on gene expression are associated with more ordered chromatin.

The manuscript is mostly well written, albeit somewhat vague and superficial in places. The findings are supported by the evidence. The main issue is that apart from the novel, rather straightforward information measure, all of the findings are expected and most also have been extensively reported previously. The other main concern I have is that the analysis is largely qualitative in nature, with simple thresholds used to include motif matches or ChIP-seq peaks to the analysis. A more quantitative approach considering several classes (high affinity, low affinity motifs, large and small ChIP-seq peaks) would be more informative.

Specific points:

- The name for the measure used, "chromatin information content", is very generic and gives the reader the wrong idea of the molecular feature that is being measured. Lack of comparison of the measure to other, more easily interpretable measures (e.g. nucleosome positioning) is also lacking.
- The measure is provided as an aggregate over a large number of loci. Distribution of the scores at individual loci would also be informative, and could be compared to peak height, motif match score etc to establish a more robust correlation between the features. Such a quantitative analysis would also help to assess whether some of the differences between the factors are caused by differential quality of the ChIP-seq experiments, or differences in distribution of the peak heights etc.

- Footprint in ATAC-seq requires high occupancy, whereas ChIP-seq can detect very low occupancy peaks. This also depends on the quality of the antibody etc. This should be clarified, and the top peaks should be analyzed separately from all peaks to see if the lack of footprints is only observed in weaker binding sites.

- The strong ordering of chromatin by CTCF is well established, and not surprising at all. CTCF tends to bind alone, whereas other transcription factors bind in clusters with many other factors. This decreases the apparent ordering of nucleosomes around them. This should at least be discussed.

- p11 the evidence is overstated. 111 out of 150 does not mean that "CIE asymmetry is intrinsic to the motif" in all cases. It only indicates that a subset of the factors have intrinsic preference

Reviewer comments are in normal black text

Our responses are in blue italic text. To enable easier re-review, we quoted text from the newly revised manuscript using left-indented blocks.

Reviewer #1 (Remarks to the Author):

The paper "Chromatin information content landscapes inform transcription factor and DNA interactions" by D'Oliveira Albanus et al. describes a novel approach to analyze and understand transcription factor binding properties by the analysis of ATAC-seq signals. The paper builds on a framework borrowed from information theory and makes large usage of a particular type of analysis (V-plots), previously conceived by the group of W. Greenleaf, to analyze global patterning of nucleosome dyads around transposase accessible sites.

Given the novelty of the approach, I would expect better characterization of the proposed metrics. Calculation of CIE and f-VICE as reported in methods remains a bit obscure, I would suggest some schematic representation (and pseudocode) to better render the underlying idea (although it can be glimpsed by Fig1A lower panel).

Response 1)

We thank the reviewer for remarking on the novelty of our approach and for suggesting a helpful schematic to clarify the methodology. We have extensively rewritten the methods subsection regarding the calculation of CIE and f-VICE, including adding equations and pseudocode for the V-plot calculation and a new schematic figure (Supplementary Fig. 1). Additionally, we revised the Methods section to improve clarity and included a new schematic figure for BMO (Supplementary Fig. 6).

In the methods section it is stated that fragments were downsampled to produce figures (Fig1b), although original values were used to perform all the analysis. This prompts me a simple question that should be addressed when CIE and f-VICE are introduced: how these values are affected by coverage in the experiment? Indeed the coverage of ATAC-seq experiments varies from 20-30M to 200M fragments, depending on the analysis scope. Since high coverage is suggested to perform footprint analysis, is this true also for information analysis? What would be the minimal and the optimal coverages for this kind of analysis? Some hints of this dependence are clear in Supplementary Figure 10 (and, in fact, depth is used as covariate to calculate normalized f-VICE).

Response 2)

We thank the reviewer for the suggestion of correlating our information metrics with sequencing coverage. To address this, we calculated f-VICEs for TF-bound motifs across downsampled datasets representing different sequencing depths. In the high-quality GM12878 ATAC-seq data generated for this study (full sequencing depth = 24 M reads), we were able to obtain f-VICE values highly correlated (Spearman's $\rho = 0.95$) to the full sample with sequencing depths as low as 5 M reads. In contrast, the GM12878 ATAC-seq dataset from Buenrostro et al (full sequencing depth = 81 M reads) required ~20 M reads to achieve 0.95 Spearman correlation.

We note, however, that even at 5 M reads, the Spearman correlation in the Buenroostro sample between the downsampled and full f-VICEs is 0.89. Therefore, we conclude that our f-VICE estimations are robust to low coverage.

These new analyses are now included in the manuscript (Supplementary Fig. 4) and we updated the text to refer to the analyses referent to this response and **Response 4**:

“The TF f-VICE values were highly concordant across GM12878 ATAC-seq libraries with distinct signal-to-noise metrics (Spearman’s ρ range 0.81-0.98, median = 0.96, $n = 7$ libraries; Supplementary Fig. 4a,b) and downsampled datasets representing sequencing depths as low as 5 M high-quality alignments (Spearman’s ρ range 0.89-0.99, median = 0.98; Supplementary Fig. 4c-e) (...) which demonstrates the robustness of the f-VICE metric.”

CIE is summarized into a single value, f-VICE. The authors should discuss if the function is bijective or not and, in case it is surjective, which V-plot configurations could lead to high f-VICE values. I expect "well organized" patterns (as CTCF) always lead to high f-VICE values, but it would be interesting to know if any other configuration, possibly more disorganized, may lead to similar values. I imagine this could be tackled theoretically and by simulating different V-plot configurations.

Response 3)

This is a very interesting point regarding the nature of the f-VICE function. The f-VICE function assumes that the the most informative regions of the CIE pattern are proximal to the feature (motif) center. Theoretically, it is possible to have different CIE configurations yielding the same f-VICE (i.e. the function is not bijective). This would be the case with features where the CIE is high at coordinates different from the ones used to calculate f-VICE. We observed that opposed nucleosome configurations at the TF binding site (i.e. well-positioned nucleosome over the motif versus phased nucleosomes adjacent to the motif) can lead to similar CIE patterns (compare AP-1 to the MAF::NFE2 and RFX motifs in the new Figure 2g). This indicates that the f-VICE calculation assumption of prominent information signals nearby the motif generally holds for different nucleosome configurations associated with TF binding. However, other genomic features could induce CIE patterns that violate this assumption. To address this issue, we performed additional analyses of the CIE patterns obtained from the features analysed in this study (TF motifs and DNA 6-mers). We show scatter plots of CIE sum versus f-VICE for all 1,793 motifs (Supplementary Fig. 28a) and 2,075 6-mers (Supplementary Fig. 28b) with sufficient coverage to calculate CIEs in our GM12878 ATAC-seq data. These analyses show the majority of sequences do not deviate from the diagonal, especially in the high f-VICE motifs/6-mers. This indicates that the majority of features do not have high CIE levels at regions different from the ones used in the f-VICE calculations. However, we noticed a subset of low f-VICE 6-mers that had higher CIE levels relative to other 6-mers in the same f-VICE range. These 6-mers had CIE patterns where the feature-distal regions had higher information

compared to the feature-proximal regions, which partially violate the f-VICE assumption (Supplementary Fig. 28b,c). Upon further analysis, we determined that this unconventional CIE shape pattern was associated with the presence of consecutive A or T nucleotides in the 6-mer sequence (Supplementary Fig. 28d). This CIE signature result is consistent with previously described A-tracts, which are poly(dA:dT) sequences that can induce bends in the DNA and are associated with nucleosome positioning (Segal & Widom, 2009 <https://dx.doi.org/10.1016%2Fj.sbi.2009.01.004>). These results indicate less common genomic features may exhibit “non-canonical” CIE patterns. We include these results as a new panel in the main text (Fig. 3i).

To additionally account for other potentially rare genomic features that could be missed by either our motif database or the 6-mers analyses, we analyzed the CIE shapes obtained from the top-scoring DNA 11-mers from gkm-SVM (Lee, 2016, <https://doi.org/10.1093/bioinformatics/btw142>) trained on our GM12878 ATAC-seq dataset. These 11-mers represent DNA sequences associated with regions of high chromatin accessibility in this cell line. We did not find any high-scoring 11-mer ($n = 1,248$) CIE patterns that diverged from the CIE patterns observed with the TF motifs (Supplementary Fig. 28f-h). This indicates that features strongly associated with regulatory element activity have the “canonical” CIE pattern we observed in TF-DNA interactions.

In summary, these results are consistent with the f-VICE equation generally capturing the observed diversity of CIE patterns. However, other less common CIE patterns are possible, potentially associated with topological properties of the DNA molecule. We therefore include a new main figure panel and its corresponding section in the manuscript, entitled “Non-canonical chromatin information shape patterns reflect local DNA topology”, related to this question:

“The f-VICE metric we developed here summarises the CIE pattern into a single value, and it was tuned to detect patterns associated with TF binding, which have higher information signal proximal to the TF motif (Fig. 1a, Supplementary Fig. 1). However, other genomic features may have distinct CIE patterns that would not be captured by the f-VICE equation. This motivated us to investigate if there is a subset of features associated with “non-canonical” V-plot patterns. We found a subset of DNA 6-mers that had higher CIE levels at regions distal to the 6-mer center (Supplementary Fig. 28a-c). These DNA 6-mers were characterized by stretches of three or more consecutive A or T nucleotides. This is consistent with these 6-mers overlapping A-tracts, which are regions characterized by poly(dA:dT) nucleotides and are associated with curvature in the DNA molecule and nucleosome organization³⁵. The CIE pattern observed in the poly(dA:dT) 6-mers results from low Tn5 integration at the 6-mer region, which was not observed in poly(dC:dG) 6-mers (Fig. 3i). In addition, we observed that the poly(dA:dT) 6-mers were generally associated with a phased nucleosome pattern in the MNase-seq data (Supplementary Fig. 28e), consistent with this DNA topology favoring nucleosome exclusion³⁴. Of note, the MNase patterns in the poly(dA:dT) 6-mers were almost indistinguishable from to the ones associated with nucleosome phasing at high f-VICE

motifs (Fig. 2g, upper panels, Supplementary Fig. 21), illustrating how the CIE metric is complementary to nucleosome positioning data. We additionally analyzed a subset of DNA 11-mers (n = 1,248) predicted to have regulatory properties in GM12878 (Methods) to determine if any of these features were associated with less common CIE patterns. We did not find evidence that these 11-mers had CIE patterns differing from the TF motifs (Supplementary Fig. 28f-h), indicating that sequences associated with regulatory activity have higher CIE levels proximal to the sequence. Together, these results demonstrate how local DNA topology is associated with higher-order chromatin organization reflected in the CIE patterns.”

This is also included in the manuscript Discussion:

“In addition, the f-VICE metric we developed here is highly tuned to detect patterns associated with TF binding, but other genomic features may have distinct CIE patterns. Indeed, we found a subset of DNA 6-mers with “non-canonical” V-plot patterns, which we hypothesized reflected a specific DNA topology (Fig. 3i, Supplementary Fig. 28). Therefore, it is reasonable to expect the existence of less common genomic features with “non-canonical” CIE patterns. One approach to systematically detect these features is through unsupervised clustering of the CIE shape patterns.”

Related to the previous two questions: how does noise influence CIE and f-VICE? There may be ATAC-seq experiments with some background noise, is the approach described robust to this?

Response 4)

*We thank the reviewer for these questions. To determine how sensitive our f-VICE metric is to dataset noise, we compared the f-VICE values calculated from bound GM12878 TF binding sites (using ENCODE ChIP-seq data) in our GM12878 ATAC-seq data to f-VICE values obtained from the same sites using all the GM12878 replicates from the Buenrostro et al study. These libraries have a range of signal-to-noise ratios, as measured by their TSS enrichment (min = 2.2, max = 5.5) and fraction of reads in peaks (min = 1%, max = 44%). Across these diverse datasets, we found that the f-VICEs were highly correlated (Spearman’s ρ range 0.81-0.98, median = 0.96, n = 7 libraries; refer to the new Supplementary Fig. 4a-b). These results, together with the downsampling analyses in **Response 2**, indicate that f-VICE calculations are robust to noise.*

*This analysis is included in the main text and is referenced in the main text excerpt in **Response 2**.*

Also, could one use the framework presented to estimate the level of noise in ATAC-seq analysis?

Response 5)

The reviewer is correct. In this manuscript, we used ubiquitous and conserved CTCF-cohesin binding sites to represent known regions of organized chromatin. By generating V-plots and comparing the CIE patterns in these regions across samples, one can visually and quantitatively determine sample quality (see Supplementary Fig. 14). As a future direction, we plan to include the CTCF-cohesin V-plot and the associated analyses in this work as a part of the ataqv ATAC-seq QC and visualization tool (<https://github.com/ParkerLab/ataqv>; Orchard et al, 2020 <https://doi.org/10.1016/j.cels.2020.02.009>).

BMO is introduced in this paper, some benchmarks were presented and discussed in the supplementary note. I would try to briefly summarize them in the main text.

Response 6)

We thank the reviewer for the helpful feedback. The Supplementary Note was made to improve the flow and readability of the manuscript by removing overly technical discussions from the main text. We revised the main text to summarize the most important benchmarking results.

“We benchmarked BMO and other TF binding prediction methods (DNase2TF12, HINT-ATAC¹³, PIQ¹¹, and CENTIPEDE¹⁹) using TF ChIP-seq data from GM12878 and HepG2 (n = 41 and n = 59, respectively; Supplementary Table 1). DNase2TF, HINT-ATAC, and PIQ rely on footprints to predict TF binding, while CENTIPEDE learns informative DNA cut patterns indicating TF binding. To compare methods, we calculated their F1 scores for predicting each TF. We additionally developed a custom implementation of CENTIPEDE that does not rely on the DNA cut patterns around the motif, which we named signal-sum CENTIPEDE (ssCENTIPEDE). ssCENTIPEDE allowed us to estimate the contribution of the DNA cut patterns around the TF motif compared to motif accessibility in CENTIPEDE predictions (Methods and Supplementary Note). BMO had overall higher performance (higher F1 score than the other methods in 74% of all comparisons; Supplementary Fig. 7). Importantly, the footprint-agnostic methods (BMO, CENTIPEDE, and ssCENTIPEDE) outperformed the footprint-based methods (higher F1 scores in 90% of comparisons), particularly on TFs with low f-VICES (Fig. 1e, Supplementary Figs. 7-11; Supplementary Note).”

The bimodal distribution of normalized f-VICE scores is, in my opinion, loosely captured by gaussian models. In supplementary figure 11 and, notably, in figure 1f, the modal value of the high-valued scores is poorly recapitulated by the gaussian distribution represented by the red line. I wonder what data have been used to build the gaussian model (possibly the data in supplementary figure 10B?). Maybe a simple Q-Q plots on the distributions displayed could be used to choose cutoffs for the two mixtures.

Response 7)

We appreciate the reviewer’s comment about the methodology used to fit the high/low f-VICE distributions. The data used to fit the distribution correspond to the values shown in previous

Supplementary Fig. 10b (currently 15b). This has been clarified and corresponding text is at the end of this response.

To further address this comment, we first fit a single Gaussian to the distributions instead of the mixture model. We found that the Bayesian information criterion (BIC) values for the mixture models were consistently greater than the single Gaussian models in all the samples analyzed in this study. This indicates that the mixture models capture the properties of the f-VICE distribution more accurately. While we agree other statistical approaches could be used to classify distributions, the main goal of this analysis was to provide a statistically motivated split between the two potential classes of motifs based on their f-VICEs, which could then be used for downstream analyses (e.g. the cis-eQTL enrichments in Fig. 3c). As suggested, we made Q-Q plots for the f-VICE values obtained from all samples analyzed in this study using either the low f-VICE fitted Gaussian distribution from the mixture model or the single fitted Gaussian as theoretical quantiles (Supplementary Fig. 16b). Note that our mixture model parameters appropriately capture the upper tail of high f-VICE motifs in the majority of samples, which can be visualized as the inflated red points at the right the plots. These new Q-Q plots indicate that the cutoffs for the two mixtures are reasonable, and we thank the reviewer for making this suggestion.

As part of the revision of the Methods section, we rewrote the related subsection to improve clarity:

“High and low f-VICE Gaussian mixture model distributions were fitted using the R package mixtools (v. 1.1.0)⁶⁰ using as input the normalized f-VICEs for each ATAC-seq sample, after filtering low signal motifs where the number of total predicted bound instances for the motif was in the lowest decile of that sample. The filtering step is to avoid potential bias for lower f-VICE values due to sparsity. We used a posterior probability of 0.5 as the threshold to split the high and low f-VICE distributions. We alternatively tried to fit a single Gaussian distribution to the f-VICEs instead of a mixture model. The Bayesian information criterion values of the Gaussian mixture models were greater than the single Gaussian models in all nine samples analyzed in this study, indicating that a mixture model was a better fit for these data. Q-Q plots show that the Gaussian mixture model and single Gaussian model thresholds can be used to obtain a similar separation between the low and high f-VICE motifs (Supplementary Fig. 16b), which indicates the robust nature of the chosen thresholds.”

There is a little discrepancy between the protocol for sonicated ATAC-seq (targeting to 350bp fragment size) and the curve in Supplementary Figure 14A, where the size distribution peaks at ~200 bp. Also, it is not clear to me if the purification step before sonication selects smaller DNA fragments (<2000 bp) or if all chromatin is eluted, although this won't probably affect results.

Response 8)

We thank the reviewer for the detailed comments about our experimental methodology. Regarding the discrepancy in the fragment size distribution, this is to be expected for two reasons. First, the sonication strength is approximate and therefore will not always result in fragment sizes that correspond to the settings used. Second, and most important, the sonication step already starts with Tn5-tagmented fragments. Therefore, the initial size distribution follows the expected size distribution from an ATAC-seq experiment (sub-nucleosomal and nucleosome-protected fragments, with lengths corresponding to ~50 and ~200 bp, respectively). The sonication step will then introduce another break to make these fragments smaller and shift the overall distribution towards smaller sizes. Given that the most of ATAC-seq fragments are already smaller than 250 bp, the sonication step will mainly shift the smaller fragments from the distribution towards smaller sizes. This leads to a fragment size distribution that is different from the conventional ATAC-seq experiment in a few aspects, as shown in the upper panel of previous Supplementary Fig. 14a (currently 19a): 1) the lack of the peak corresponding to sub-nucleosomal fragments (fragment length ~50 bp), which are likely destroyed in the sonication process; 2) a mononucleosome peak (fragment length ~200 bp) that is higher and extends further to the left (to ~100 bp) in the sonicated sample compared to the conventional sample.

Regarding the column purification step, it is possible that very large fragments could be selected against. However, as the reviewer correctly indicated, these larger fragments contribute very little to our chromatin information metric. The majority of the chromatin information signal originates from the sub-nucleosomal and nucleosome-sized fragments near the TF binding site. To further address this comment, we generated V-plots and calculated f-VICEs for our non-sonicated sample excluding fragments greater than 1 kb. As seen in the new Supplementary Fig. 19c, these f-VICE values are highly correlated (Spearman's $\rho = 0.99$) with the values from the full fragment length distribution sample. This indicates that any potential experimental bias that would select against larger fragments will have negligible effects on our chromatin information metrics. This is also included in the main text:

“There were no detectable nucleosome phasing patterns in the motif-flanking CIE signature (~50 bp from the motif) of the sonicated sample (vertical arrows in Fig. 2d and Supplementary Fig. 19b), which we determined was not due to size-selection bias in the library preparation (Supplementary Fig. 19c).”

When testing TSS-distal/proximal CIE asymmetry a binomial test is used. While 111/150 is an impressive fraction and I'm persuaded it is a good result, a better test would be a fisher test for distal/proximal and agree/disagree, especially if the intended scope of the test is "To determine if asymmetric CIE was an artifact of TSS proximity".

Response 9)

We thank the reviewer for the positive evaluation of our CIE asymmetry results and for the suggestion to use an additional statistical test. We performed a Fisher test as per the reviewer's suggestion, using the observed versus expected agreement under the null hypothesis (50/50

ratio). The test was significant ($p = 2.8e-5$) and we now include the results for the binomial test and Fisher's in the revised manuscript.

Lastly, I have a more general comment on the methods presented in this paper: once the list of genomic intervals for ATAC sites is computed, they can be iteratively sampled to build V-plots and calculate CIE/f-VICE. The resulting scores will distribute in a multimodal way, as expected from the results presented. If the sampling strategy is opportunely tuned, this procedure should tend to a distribution of scores representative of all TF that could bind chromatin (in theory) or at least to some major clusters (e.g. low-CIE, mid-CIE, high-CIE). Do the authors think such approach could be useful and applied to study chromatin properties in non-model organisms? If yes, I suggest to add a comment in the discussion.

Response 10)

The reviewer brings an excellent point that warrants further discussion. It is feasible to calculate V-plots and f-VICEs in other organisms using any set of genomic features. A DNA k-mer based approach similar to the one used for this study would allow one to determine the subset of DNA sequences associated with the highest degree of chromatin organization in any organism, completely independent of any previous knowledge about the TF repertoire of that organism (although previous analyses suggests that TF DNA binding preferences are highly conserved throughout the eukaryotic tree (Wilson et al, 2008 <https://doi.org/10.1126/science.1160930>; Nitta et al, 2017 <https://doi.org/10.7554/eLife.04837>; Wong et al 2019 <https://doi.org/10.1126/science.1160930>).

Regarding the understanding of TF biology, we think an exciting application of the reviewer's suggestion is to calculate the f-VICE distribution of DNA k-mers using ATAC-seq data from organisms across the eukaryotic tree. Based on when the emergence of the bi-modal f-VICE distribution occurs in eukaryotes, one could reconstruct the evolutionary history of TFs with nucleosome-phasing properties (putative pioneer TFs) and their impact on genome organization.

We updated the manuscript to include this topic in the discussion:

“An exciting potential application of the methodology presented here is to estimate chromatin information patterns in other organisms, including non-model organisms where less information about their TF repertoire is available. We reason that the unbiased estimation of information patterns encoded in DNA substrings occurring in accessible chromatin, such as we performed using DNA 6-mers (Fig. 3e), can be used to inform the possible chromatin organization configurations associated with that organism. Such an approach could potentially be used to determine the appearance of pioneer-like TFs along the eukaryotic tree, which would be reflected in the emergence of a long right tail in the f-VICE distribution (Fig. 3e and Supplementary Fig. 25a).”

Minor comments:

- if BMO is an acronym, it is never explained in the text

Response 11)

BMO means “bee model of TF binding”, corresponding to the analogy of TFs behaving as “Brownian bees” in the genome looking for TF binding sites (“flowers”). The more accessible and the greater quantity of flowers, the more likely the bee will randomly interact with them and remain in the region. This explanation is now included in the main text and Methods.

- page 7, fig 1g is mentioned but the panel is missing from the figure and its legend

Response 12)

We apologize for the typo; it should read Fig. 1f. This is now fixed in the current version.

- I suggest a general revision of methods to improve clarity.

Response 13)

We thank the reviewer for the suggestion. We extensively revised the methods section of the manuscript to improve clarity.

Reviewer #2 (Remarks to the Author):

In the manuscript entitled "Chromatin information content landscapes inform transcription factor and DNA interactions" Albanus et al. devise a metric for chromatin organization at specific positions of the genome that they term "chromatin information content", and use it to classify different transcription factor ChIP-seq peaks, motif matches and 6 bp subsequences. They find that features such as higher residency time of transcription factor and genetic effects on gene expression are associated with more ordered chromatin.

The manuscript is mostly well written, albeit somewhat vague and superficial in places. The findings are supported by the evidence. The main issue is that apart from the novel, rather straightforward information measure, all of the findings are expected and most also have been extensively reported previously.

Response 14)

We thank the reviewer for the overall positive assessment of our manuscript. Here, we clarify the results that are novel contributions to the field versus those that we consider complementary and reinforcing to existing publications. Regarding the novel contributions, we developed a new information theory method to quantify chromatin organization, called chromatin information enrichment (CIE). CIE is more general and complementary to analyzing nucleosome positioning, as it captures additional features such as DNA protection from TF binding and local DNA topology (see the new Fig. 3i). We show that features such as high TF residence time and genetic regulation of gene expression are associated with highly ordered chromatin. To our knowledge this is the first time that these features are linked together. A previous study from Gordon Hager's group raised the possibility that TF residence time affects chromatin

accessibility patterns (Sung et al, <http://dx.doi.org/10.1016/j.molcel.2014.08.016>). However, this was shown based on previously reported residence times for only three TFs and focused on DNase-seq footprinting depth, a metric that solely reflects DNA protection at the motif vicinity and is therefore different to our information theory approach. In this study, we quantify chromatin organization patterns of hundreds of TFs and use previously reported residence times from 11 TFs to demonstrate that residence time is associated with chromatin organization. In addition, we estimate that only 10-20% of TFs associate with highly organized chromatin, likely due to their high residence times. We then show that this subset of TFs are more highly enriched to overlap genetic variants associated with the control of gene expression and chromatin accessibility, which we hypothesize is due to pioneer-like properties of these TFs. To our knowledge, there are no other studies that performed a similar analysis of TF-chromatin interaction patterns across human cell lines and tissues. Therefore, we believe that our work brings novelty to the field of chromatin biology and gene regulation.

We additionally provide strong evidence that footprinting-based methods are not an optimal strategy to predict TF binding. Despite Gordon Hager's group and others having shown the limitations of footprinting (Baek et al, 2017 <http://dx.doi.org/10.1016/j.celrep.2017.05.003>), footprinting-based algorithms are still being used and actively developed. We believe that the findings described here will help the field move towards new TF binding prediction strategies that rely on other aspects of TF-chromatin interactions or that specifically take into account the biological properties of the TFs of interest, such as their residence times.

To address this comment, we rewrote sections of the discussion to highlight the novelty of our findings:

“In this study, we develop and use for the first time, entropy-based algorithms to analyze chromatin accessibility data and quantify the level of chromatin organization at genomic features of interest. This chromatin information approach is more general and complementary to analyzing nucleosome positioning, as it captures additional features such as DNA protection from TF binding and local DNA topology. We use this entropy-based approach to dissect TF-chromatin interaction patterns across human cell lines and tissues. The TF-chromatin interactions are captured in the information content patterns of chromatin accessibility and reflect functional properties of TFs, such as TF-DNA residence times, specific protein domains, and the ability to induce nucleosome repositioning. We find that a subset of TFs (10-20%) have high chromatin information and are more highly associated with the genetic control of both chromatin accessibility and gene expression. We hypothesize that these TFs define cell state by potentially acting as pioneers.”

“Finally, we show that footprinting-based algorithms to predict TF binding, which remain a popular choice in the field, are sensitive to the TF-chromatin information landscape we describe. We develop and cross-validate a novel tool for predicting TF binding based on

chromatin accessibility that outperforms footprinting-based methods. These findings represent strong evidence that most TF binding sites do not associate with footprints.”

The other main concern I have is that the analysis is largely qualitative in nature, with simple thresholds used to include motif matches or ChIP-seq peaks to the analysis. A more quantitative approach considering several classes (high affinity, low affinity motifs, large and small ChIP-seq peaks) would be more informative.

Response 15)

*We thank the reviewer for their comment about the granularity in the thresholds used for our analyses, as we agree this can be improved. We reanalyzed the data using a more quantitative approach to measure chromatin information based on different classes, as suggested: ATAC-seq signal intensity, ChIP-seq signal intensity, and predicted TF-DNA thermodynamic affinity (Stormo, 2000 <https://doi.org/10.1093/bioinformatics/16.1.16>; Zhao and Stormo, 2011 <https://doi.org/10.1038/nbt.1893>). These analyses are described in more detail in related **Response 18**, below. In summary, we find that these new more quantitative analyses support our previous findings and indicate robustness of the thresholds used to separate motif classes.*

Specific points:

- The name for the measure used, "chromatin information content", is very generic and gives the reader the wrong idea of the molecular feature that is being measured.

Response 16)

We thank the reviewer for this helpful feedback. We purposely aimed for a generic name given that this approach can be used in a generic way. To illustrate this, we performed CIE calculations at transcription start sites (TSS) from highly expressed genes in GM12878. The new Supplementary Fig. 29 shows that the V-plots can be used to visualize the asymmetric ATAC-seq patterns around the TSS, including a higher number of nucleosome-sized fragments immediately downstream of the TSS consistent with a well-positioned +1 nucleosome. This is now included in the manuscript:

“While we focused this study on TF-chromatin interactions, the CIE framework presented here can be used to study other genomic features. To illustrate this, we generated V-plots and calculated CIE for the TSS regions from highly expressed in GM12878. Using this approach, we can observe the highly asymmetrical accessibility pattern in the TSS, indicating a well-positioned +1 nucleosome downstream of the TSS (Supplementary Fig. 29). This demonstrates the versatility of our entropy-based methodology to characterize genomic features which would otherwise require laborious experimental approaches or would not be possible in vivo.”

*In addition, we believe that more details about the CIE metric can help clarify what is being measured. Please refer to **Response 17** below for a revised description and interpretation of the CIE results.*

Lack of comparison of the measure to other, more easily interpretable measures (e.g. nucleosome positioning) is also lacking.

Response 17)

We thank the reviewer for this comment. Our CIE metric measures the overall level of chromatin organization at genomic features of interest, which is reflected in the ATAC-seq fragment patterns, and is not a specific nucleosome-centric metric, even though it can capture these patterns.

*We performed a direct comparison between nucleosome positioning and CIE patterns (please see new panels e-g in Figure 2 and the new Supplementary Figs. 20,21). We also repeated these analyses using nucleosome positioning results directly obtained from the ATAC-seq data using the NucleoATAC algorithm (Schep et al, 2015, <https://doi.org/10.1101/gr.192294.115>) (Supplementary Fig. 20c-e). Based on these analyses we find that high f-VICE TFs generally associate with nucleosome phasing adjacent to the motif region, consistent with their predicted (and measured) high residence times. However, approximately 20% of the high f-VICE TFs associate with a well-positioned nucleosome directly over the motif region, which could result from the TF binding and stabilizing a nucleosome. This indicates that high CIE reflects two distinct nucleosomal organizations and supports the generalized property of this approach. (Fig. 2f,g and Supplementary Fig. 21). Further supporting this result, the CIE metric allowed us to identify signatures of a specific A-tract DNA topology (detailed in **Response 3**). Therefore, the CIE methodology presented here is more general and complementary to analyzing nucleosome positioning.*

We rewrote the section of the manuscript corresponding to these analyses to highlight these differences:

“To systematically characterize the association between CIE and nucleosome positioning, we compared GM12878 CIE patterns across TF motifs to the nucleosome positions obtained both from ATAC-seq using the NucleoATAC algorithm²⁶ and from lymphoblastoid MNase-seq profiles (Supplementary Fig. 20). High f-VICE motifs had lower nucleosome occupancy directly at the motif region and phased nucleosomes directly adjacent to it (Fig 2e, Supplementary Fig. 20e). Accordingly, the CIE patterns of high f-VICE motifs were significantly more likely to be anti-correlated with the MNase-seq signal at the motif region ($p = 2e-13$, generalized linear model; Fig. 2f,g, upper two panels, Supplementary Fig. 20f). We calculated the degree of nucleosome phasing around the motif region and found that it was significantly correlated with f-VICE (Pearson’s $\rho = 0.4$, $p = 4e-22$; Supplementary Fig. 20g). However, we observed that 22% of the high f-VICE motifs had high MNase signal at the motif region (12/54; Fig. 2g,

bottom two panels, Supplementary Fig 21). This indicates that high CIE patterns more commonly result from nucleosome phasing induced by TF binding, but can also result from a well-positioned nucleosome at the TF binding site. The latter case is consistent with TFs that bind at the nucleosome dyad^{4,27}, which include a member of the RFX family⁴. These results indicate that the CIE levels reflect the overall level of chromatin organization and can capture different nucleosomal configurations. Therefore, the CIE patterns are more general and complementary to nucleosome positioning data.”

- The measure is provided as an aggregate over a large number of loci. Distribution of the scores at individual loci would also be informative, and could be compared to peak height, motif match score etc to establish a more robust correlation between the features. Such a quantitative analysis would also help to assess whether some of the differences between the factors are caused by differential quality of the ChIP-seq experiments, or differences in distribution of the peak heights etc.

Response 18)

We thank the reviewer for encouraging us to more quantitatively deconvolute the f-VICE signal properties. To address these comments, we recalculated f-VICEs on all bound motifs after splitting them in quantiles of ChIP-seq signal, ATAC-seq signal, and TF affinity (calculated using equation 4 from Stormo, 2000, <https://doi.org/10.1093/bioinformatics/16.1.16>), which is a proxy of the thermodynamic strength of TF-DNA interactions (Zhao and Stormo 2011 <https://doi.org/10.1038/nbt.1893>). We used a regression model to quantify the effects of each parameter on f-VICE values. As the new Supplementary Fig. 5 demonstrates, the f-VICE signal is stable across all subsets of motifs and is highly correlated to the f-VICEs calculated from all bound motif instances (Spearman's ρ range 0.7-0.89, median = 0.85). Based on these results, we conclude that our f-VICE metric robustly captures the properties of the TF-DNA interactions.

We included these updated analyses in the main text:

“We additionally calculated f-VICE values separately for motif instances across quintiles of ChIP-seq signal intensity, chromatin accessibility, and TF affinity. We found that the f-VICE values were highly correlated across the motif subsets (Spearman's ρ range 0.7-0.89, median = 0.85; Supplementary Fig. 5a-b). We performed a linear regression of f-VICE controlling for these potentially confounding metrics and found that the model coefficients for the TF terms were highly correlated with their respective f-VICEs (Spearman's $\rho = 0.9$, $p < 1e-323$; Supplementary Fig. 5c), which demonstrates the robustness of the f-VICE metric.”

- Footprint in ATAC-seq requires high occupancy, whereas ChIP-seq can detect very low occupancy peaks. This also depends on the quality of the antibody etc. This should be clarified, and the top peaks should be analyzed separately from all peaks to see if the lack of footprints is only observed in weaker binding sites.

Response 19)

We thank the reviewer for this comment. We analyzed the motifs intersecting the top 20% ChIP-seq peaks separately and we found that one of the footprinting-based methods, DNase2TF, had slightly higher performance on a subset of TFs compared to BMO only in our higher signal-to-noise GM12878 ATAC-seq dataset. DNase2TF had lower performance when using the lower signal-to-noise Buenrostro et al. GM12878 data. These lower signal-to-noise quality control metrics (TSS enrichment and fraction of reads in ATAC-seq peaks) compared to our GM12878 data shows the performance of DNase2TF is sensitive to sample quality. Importantly, the slightly increased performance was not observed for any of the low f-VICE TFs, indicating that even at high occupancy sites, low f-VICE TFs do not associate with footprints. These results indicate that footprinting-based approaches are useful only for a small subset of TFs and TF binding sites. These analyses are now included in the new Supplementary Fig. 12.

We included this analysis in the main text:

“To determine if the overall lower performance of the footprinting-based methods resulted from only sites with strong TF affinity being associated with footprints, we benchmarked all the methods using TF binding sites in the top and bottom 20% of ChIP-seq signal defined TF occupancy separately (Supplementary Figure 12) . The footprinting-based method DNase2TF had higher performance compared to BMO when predicting the top 20% binding sites (Supplementary Figure 12a). However, the increase in performance for DNase2TF was limited to medium and high f-VICE TFs and only when using the high signal-to-noise GM12878 ATAC-seq dataset generated for this study (Supplementary Fig. 12a, left). BMO outperformed DNase2TF in the top 20% binding sites when using the Buenrostro et al. GM12878 dataset⁷ (Supplementary Fig. 12a, right), which had a lower signal-to-noise ratio (Supplementary Fig. 2). These results are consistent with only a subset of high-occupancy binding sites from high f-VICE TFs associating with detectable footprints and that footprint detection is sensitive to sample quality.”

- The strong ordering of chromatin by CTCF is well established, and not surprising at all. CTCF tends to bind alone, whereas other transcription factors bind in clusters with many other factors. This decreases the apparent ordering of nucleosomes around them. This should at least be discussed.

Response 20)

We thank the reviewer for bringing up this important point. We added text to the discussion to address this comment:

“One limitation of our methodology is that it can be affected by clusters of TFs which can potentially decrease the apparent information of the local chromatin. This can be circumvented with careful experimental approaches to separate these TF binding sites, such as the one we used for CTCF and cohesin.”

- p11 the evidence is overstated. 111 out of 150 does not mean that "CIE asymmetry is intrinsic to the motif" in all cases. It only indicates that a subset of the factors have intrinsic preference

Response 21)

We thank the reviewer for correctly pointing out our ambiguous phrasing. We meant to say that a subset of TFs have asymmetric binding, as the reviewer suggests. We rephrased the statement to improve clarity:

"These results indicate that a subset of TFs are associated with asymmetric TF-chromatin interactions."

References

- Baek S., I. Goldstein, and G. L. Hager, 2017 Bivariate Genomic Footprinting Detects Changes in Transcription Factor Activity. *Cell Reports* 19: 1710–1722. <https://doi.org/10.1016/j.celrep.2017.05.003>
- Lee D., 2016 LS-GKM: a new gkm-SVM for large-scale datasets. *Bioinformatics* 32: 2196–2198. <https://doi.org/10.1093/bioinformatics/btw142>
- Nitta K. R., A. Jolma, Y. Yin, E. Morgunova, T. Kivioja, *et al.*, 2015 Conservation of transcription factor binding specificities across 600 million years of bilateria evolution. *Elife* 4. <https://doi.org/10.7554/eLife.04837>
- Orchard P., Y. Kyono, J. Hensley, J. O. Kitzman, and S. C. J. Parker, 2020 Quantification, Dynamic Visualization, and Validation of Bias in ATAC-Seq Data with *ataqv*. *Cell Systems* 10: 298-306.e4. <https://doi.org/10.1016/j.cels.2020.02.009>
- Schep A. N., J. D. Buenrostro, S. K. Denny, K. Schwartz, G. Sherlock, *et al.*, 2015 Structured nucleosome fingerprints enable high-resolution mapping of chromatin architecture within regulatory regions. *Genome Res.* 25: 1757–1770. <https://doi.org/10.1101/gr.192294.115>
- Segal E., and J. Widom, 2009 Poly(dA:dT) tracts: major determinants of nucleosome organization. *Current Opinion in Structural Biology* 19: 65–71. <https://doi.org/10.1016/j.sbi.2009.01.004>

Sung M. H., M. J. Guertin, S. Baek, and G. L. Hager, 2014 DNase footprint signatures are dictated by factor dynamics and DNA sequence. *Molecular Cell* 56: 275–285. <https://doi.org/10.1016/j.molcel.2014.08.016>

Stormo G. D., 2000 DNA binding sites: representation and discovery. *Bioinformatics* 16: 16–23. <https://doi.org/10.1093/bioinformatics/16.1.16>

Wilson M. D., N. L. Barbosa-Morais, D. Schmidt, C. M. Conboy, L. Vanes, *et al.*, 2008 Species-Specific Transcription in Mice Carrying Human Chromosome 21. *Science* 322: 434–438. <https://doi.org/10.1126/science.1160930>

Wong E. S., S. Z. Tan, V. Garside, G. Vanwalleghem, F. Gaiti, *et al.*, 2019 *Early origin and deep conservation of enhancers in animals*. *Evolutionary Biology*.

Zhao Y., and G. D. Stormo, 2011 Quantitative analysis demonstrates most transcription factors require only simple models of specificity. *Nat Biotechnol* 29: 480–483. <https://doi.org/10.1038/nbt.1893>

REVIEWERS' COMMENTS

Reviewer #1 (Remarks to the Author):

The reviewed manuscript is richer and clearer, a couple of minor points remains to be addressed, though.

- Supplementary figure 5c: if f-VICE is the dependent variable in the model, the axes of the scatterplot should be swapped
- Careful review of references to figure numbers should be performed, for example at page 13, line 291, figure S20c-e is referenced whereas it should be S25c-e (I believe)
- Reading the response I realized p-values are reported without decimal digits (e.g. the p of Fisher's test for assimetry is $3e-5$ in the text and $2.8e-5$ in the response). Please update the values with at least 2 decimal digits.

REVIEWERS' COMMENTS

Reviewer #1 (Remarks to the Author):

The reviewed manuscript is richer and clearer, a couple of minor points remains to be addressed, though.

We kindly thank the reviewer for the positive assessment of our revisions.

- Supplementary figure 5c: if f-VICE is the dependent variable in the model, the axes of the scatterplot should be swapped

The reviewer is correct. We swapped the axes of Supp. Fig. 5c accordingly.

- Careful review of references to figure numbers should be performed, for example at page 13, line 291, figure S20c-e is referenced whereas it should be S25c-e (I believe)

We thank the reviewer for the careful evaluation of our manuscript. We corrected this mislabeled figure reference.

- Reading the response I realized p-values are reported without decimal digits (e.g. the p of Fisher's test for asymmetry is $3e-5$ in the text and $2.8e-5$ in the response). Please update the values with at least 2 decimal digits.

We recorded three significant digits for all p values in the manuscript.